# Innate lymphocyte-induced CXCR3B-mediated melanocyte apoptosis is a potential initiator of T-cell autoreactivity in vitiligo

Meri K. Tulic [1], Elisa Cavazza[1], Yann Cheli [2], Arnaud Jacquel [3], Carmelo Luci [4], Nathalie Cardot-Leccia[5], Hanene Hadhiri-Bzioueche[1], Patricia Abbe[1], Maéva Gesson [6], Laura Sormani[1], Claire Regazzetti[1], Guillaume E. Beranger [1], Cedric Lereverend [2], Caroline Pons[1], Abdallah Khemis[7], Robert Ballotti[2], Corine Bertolotto[2], Stéphane Rocchi[1] & Thierry Passeron [1,7]

T-cells play a crucial role in progression of autoimmunity, including vitiligo, yet the initial steps triggering their activation and tissue damage remain unknown. Here we demonstrate increased presence of type-1 innate lymphoid cells (NK and ILC1)-producing interferon gamma (IFNγ) in the blood and in non-lesional skin of vitiligo patients. Melanocytes of vitiligo patients have strong basal expression of chemokine-receptor-3 (CXCR3) isoform B which is directly regulated by IFNγ. CXCR3B activation by CXCL10 at the surface of cultured human melanocytes induces their apoptosis. The remaining melanocytes, activated by the IFNγ production, express co-stimulatory markers which trigger T-cell proliferation and subsequent anti-melanocytic immunity. Inhibiting the CXCR3B activation prevents this apoptosis and the further activation of T cells. Our results emphasize the key role of CXCR3B in apoptosis of melanocytes and identify CXCR3B as a potential target to prevent and to treat vitiligo by acting at the early stages of melanocyte destruction.

[1] Team 12, Université Côte d'Azur, INSERM U1065, Centre Méditerranéen de Médecine Moléculaire (C3M), Nice 06200, France. [2] Team 1, Université Côte d'Azur, INSERM U1065, Centre Méditerranéen de Médecine Moléculaire (C3M), Nice 06200, France. [3] Team 2, Université Côte d'Azur, INSERM U1065, Centre Méditerranéen de Médecine Moléculaire (C3M), Nice 06200, France. [4] Team 8, Université Côte d'Azur, INSERM U1065, Centre Méditerranéen de Médecine Moléculaire (C3M), Nice 06200, France. [5] Côte d'Azur University, Department of Pathology, CHU Nice, Nice 06200, France. [6] Imaging Facility, Nice 06200, France. [7] Côte d'Azur University. Department of Dermatology, CHU Nice, Nice 06200, France. Correspondence and requests for materials should be addressed to M.K.T. (email: meri.tulic@unice.fr) or to T.P. (email: passeron@unice.fr)

During the past decade significant advances in the understanding of vitiligo pathophysiology have been made. Large GWAS studies showed the role of allelic variations in genes involved in melanogenesis and immunity in vitiligo patients[1–3]. The development of mouse models provided very useful clues on the implication of the adaptive immune system in the disappearance of melanocytes in the skin and hair follicles[4–7]. Recent data emphasized the key role of the interferon gamma (IFNγ) pathway in vitiligo. It has been recently shown that under IFNγ stimulation, keratinocytes are stimulated to produce chemokines, such as CXCL9 or CXCL10, that further attract and activate CD8+ T cells[8]. Chemokine (C-X-C motif) receptor 3 (CXCR3) is the main receptor for these chemokines and it has been reported to be increased in vitiligo patients[9]. Targeting total CXCR3 by using depleting antibodies has provided encouraging results in vitiligo mouse model[10]. However, the initial source of IFNγ remains obscure. Using transcriptional analysis, we recently demonstrated a significant increase in CXCL10 in the non-lesional (NL) vitiligo skin compared to healthy controls[11]. Significant increases were also seen for natural killer (NK) receptors including NKTR and KLRC1 as well as trends for increased EOMES (master regulator of NK cells), CCL20 and NK-related cytokines (TNFα and IL-15) which were all increased at least 1.5-fold but did not reach statistical significance[11]. These data are in accordance with the first transcriptome analysis comparing gene expression profiles of skin from vitiligo patients with normal skin of healthy volunteers that showed an increased expression of genes associated with innate immune responses, particularly NK cell function, activity and cytoxicity[12]. These included multiple NK cell activation markers such as KLRK1 (also known as NKG2D), KLRC2 and KLRC4, ligands for NK receptor (CLEC2B), as well as markers of oxidative stress (CANP and POSTN) and innate immunity (DEFB103A). Interestingly, all of these genes were also upregulated in the NL skin of vitiligo patients, suggesting that the activation of the innate immunity may be present throughout the entire skin surface of vitiligo patients.

NK cells are part of the group 1 innate lymphoid cells (ILC1) that act as first line defense against microbial and viral infections, early cellular transformation and tumour growth[13]. NK cells have potent cytotoxic function and are able to discriminate and eliminate stressed-cells because of a large panel of activating and inhibitory receptors. Furthermore, NK cells also produce a wide array of cytokines and chemokines to recruit and instruct other immune cells for subsequent priming (adaptive response)[14,15]. For this reason, NK cells are described as a bridge between innate and adaptive immune system. It's been known for almost 30 years that there is an increase in circulating NK cells in the blood of vitiligo patients[16], yet their role in vitiligo skin remains largely unexplored. In addition to NK cells, the ILC family encompasses ILC1, ILC2, ILC3 and ILC regulatory subclasses. Similarities between ILCs subsets have led to suggestions that ILCs are the innate counterparts of T cell subsets. These are equivalent to adaptive immune system T-helper (Th)1, Th2, Th17 and Treg-like, respectively. ILCs are scarce cells in normal human skin but are upregulated in atopic dermatitis (ILC2) and psoriasis skin (ILC3)[17]. Latest research suggests that ILCs are highly heterogeneous cell types between individuals and tissues and they can even convert to a different subtype during inflammation and disease (eg., ILC3 converting to ILC1 in colorectal cancer)[18].

Together these data suggest that NK cells and innate immune system may play an important role in the early stages of vitiligo which may begin in normal-appearing, NL skin of the patient. On the other hand, it has been demonstrated that melanocytes from vitiligo patients have intrinsic defects[19]. Moreover, there have been some clues in the literature from work published over 20 years ago showing that melanocytes themselves can present antigens[20,21]. Clinical and histological data showing a restrained immunity against melanocytes in halo nevi[22] or in localized depigmentation of genital melanosis[23] strongly suggest the key role of melanocytes in initiating the anti-melanocytic immunity. However, the exact role of melanocytes in trigerring the initial steps of auto-immunity in vitiligo is far from being fully understood.

We hypothesized that vitiligo is initiated by dysregulation in innate immunity and melanocytes, and that innate immune cells are the initial source of IFNγ production in the skin. Due to their potent IFNγ producing capacity, we chose to focus on NK/ILC1 in this study. Our objective was to assess the role of the innate immunity and melanocytes in the initial steps of vitiligo by using melanocytes and blood extracted from vitiligo patients compared to healthy volunteers. Here we show data suggesting that vitiligo is likely to be initiated in the normal-appearing skin of vitiligo subjects through activation of local innate lymphocyte-induced melanocyte apoptosis and subsequent activation of adaptive and memory immune responses. This apoptosis is mediated by melanocytic CXCR3B, identifying this isoform of CXCR3 chemokine receptor to be critical in melanocyte destruction and initiation of the disease.

## Results

**Increased NK and ILC1 in vitiligo skin and blood**. To characterise the innate immune cells present in the skin and blood of vitiligo patients and to examine whether these cells were the source of IFNγ, we used specific antibodies directed against NK and ILC1 subpopulation of innate immune cells and examined their presence in vitiligo patients compared to healthy controls[24]. Characteristics of vitiligo and control populations are presented in Table 1. Our immunofluorescence results have shown that indeed, cytotoxic NK cells (defined as CD56 + Granzyme B+) containing IFNγ (CD56 + IFNγ+) were present in NL human vitiligo skin (Fig. 1a, b). Their presence was confirmed by immunohistochemistry, and these cells were located just under the epidermis of NL but not in lesional (L) skin (Fig. 1c). Semi-quantification of their detection has shown significantly higher

**Table 1 Patient characteristics**

| Characteristics | Healthy | Vitiligo |
|---|---|---|
| Subjects (*n*) | 8 | 14 |
| Age (years), median (IQR) | 51 (39–74) | 48 (43–56) |
| Gender (male/female) | 5/3 | 7/7 |
| Disease duration | – | 4 months–14 years |
| Family history (%) | – | 1 (7%) |
| Auto-immune thyroid disease (%) | – | 2 (14%) |
| Location of biopsy (*n*) | Inner arm (3), leg (2), back (1), abdomen (1) and buttock (1) | Inner arm (5), leg (3), back (2), abdomen (2) and buttock (2) |

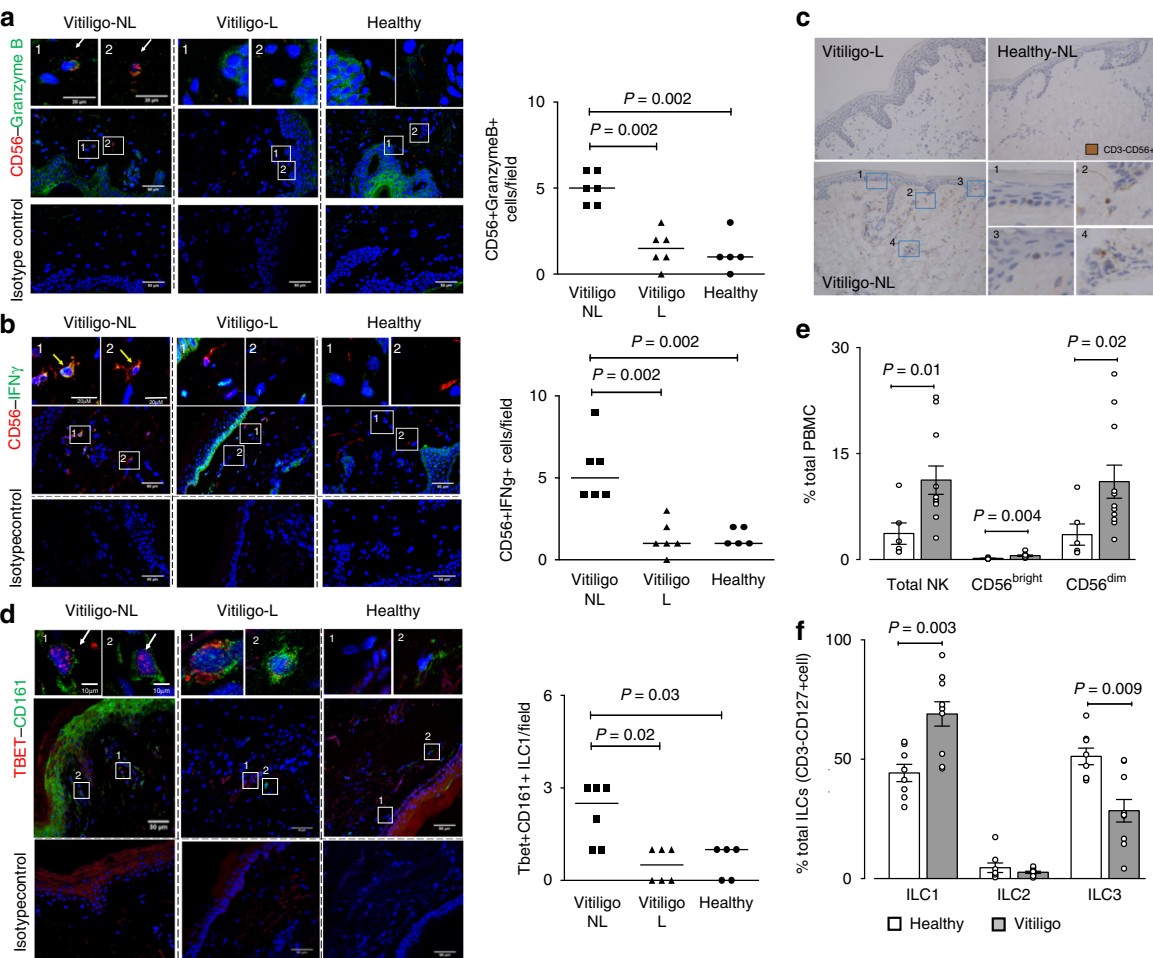

**Fig. 1** Increased detection of innate immune cells in non-lesional skin of vitiligo patients. Immunofluorescence depicting NK cells (**a**, **b**, **c**) and ILC1 (**d**, **e**) in the skin and in the peripheral blood (**f**, **g**) of vitiligo patients ($n = 6$) and healthy controls ($n = 5$). Number of yellow-immunoreactive CD56 + GranzymeB+ NK cells and their semi-quantification in non-lesional (NL) and lesional (L) skin of vitiligo patient compared to healthy skin are shown in **a** and the number of CD56+IFNγ+NK in **b**. The presence of NK cells (CD3-CD56+) was confirmed by immunohistochemistry showing cells to be located just under the epidermis of non-lesional (NL) skin (**c** bottom left and magnified in 1–4) but less in lesional (L) (**c** top) or healthy skin (**c** top right). **d** Depicts Tbet+CD161 +ILCs in the skin and their quantification. In the peripheral blood total NK and separately, cytokine producing NKs (CD56[bright]) and cytotoxic NKs (CD56[dim]) were examined in 6 healthy and 11 vitiligo patients and results represented as % of total PBMC (**e**). The percentage of ILC1, ILC2 and ILC3 in the blood of healthy ($n = 8$) and vitiligo ($n = 10$) patients is shown in **f**. Results are shown as individual dot plots with a line at median (**a**–**d**) or as means ± SEM (**e**, **f**)

number of CD56 + Granzyme B+ cells and CD56 + IFNγ+cells in NL compared to L skin of vitiligo patients ($n = 6$) and a higher number in NL skin of vitiligo patients compared to healthy skin ($n = 5$) (Fig. 1a, b). We have also detected presence of Tbet + CD161+ ILC1 in vitiligo skin (Fig. 1d) which were confirmed to be negative for CD3 and CD56 and positive for IL-7Rα CD127. Like NK cells, ILC1 cells were significantly more detected in NL vitiligo skin compared to L vitiligo skin or to healthy controls (Fig. 1d).

Next, we went on to examine whether similar differences in the number of innate immune cells was seen in the peripheral blood. Total NK cells make up ~1–5% of total PBMC in healthy subjects (Fig. 1e). The majority of NK cells were CD56[dim], that is, the cytotoxic phenotype (~90%). Total NK population (CD3-CD56+), cytokine-producing CD56[bright] NK cells as well as cytotoxic CD56[dim] NK populations were significantly increased in the blood of vitiligo patients ($n = 11$) compared to healthy controls ($n = 6$) (Fig. 1e and Supplementary Fig. 1a). The higher increase was observed for CD56[bright] NK cells. ILCs are rare cells making up < 0.1% of total PBMC. In healthy subjects, ILC1 and ILC3 (defined

as Lin-CD127 + CRTh2-CD117−  and Lin-CD127+CRTh2-CD117+cells, respectively), each made up 30–40% of total ILC population, with ILC2 (defined as Lin-CD127+CRTh2 + CD117+ cells) making up the rest (Fig. 1f and Supplementary Fig. 1b). We have shown the proportion of ILC1 in PBMC of vitiligo subjects ($n = 10$) to be significantly increased compared to controls ($n = 8$) while the proportion of ILC3 was significantly decreased and ILC2 remained unchanged (Fig. 1f).Together, these results have shown that there is increased presence of NK and ILC1 in the NL skin of vitiligo patients and in their peripheral blood.

**Stressed NK/ILC1 produce IFNγ and induce melanocytic chemokines.** Hydrogen peroxide can damage DNA, lipids, and proteins and induces damage-associated molecular pattern (DAMP) molecules that are one of the main activating factors for innate cells. It is well demonstrated that vitiligo melanocytes possess higher susceptibility to oxidative insults[25,26]. We sorted NK and total ILCs from PBMC of healthy (C) and vitiligo (V) patients and examined their IFNγ responses 24, 48 and 72 h post stimulation

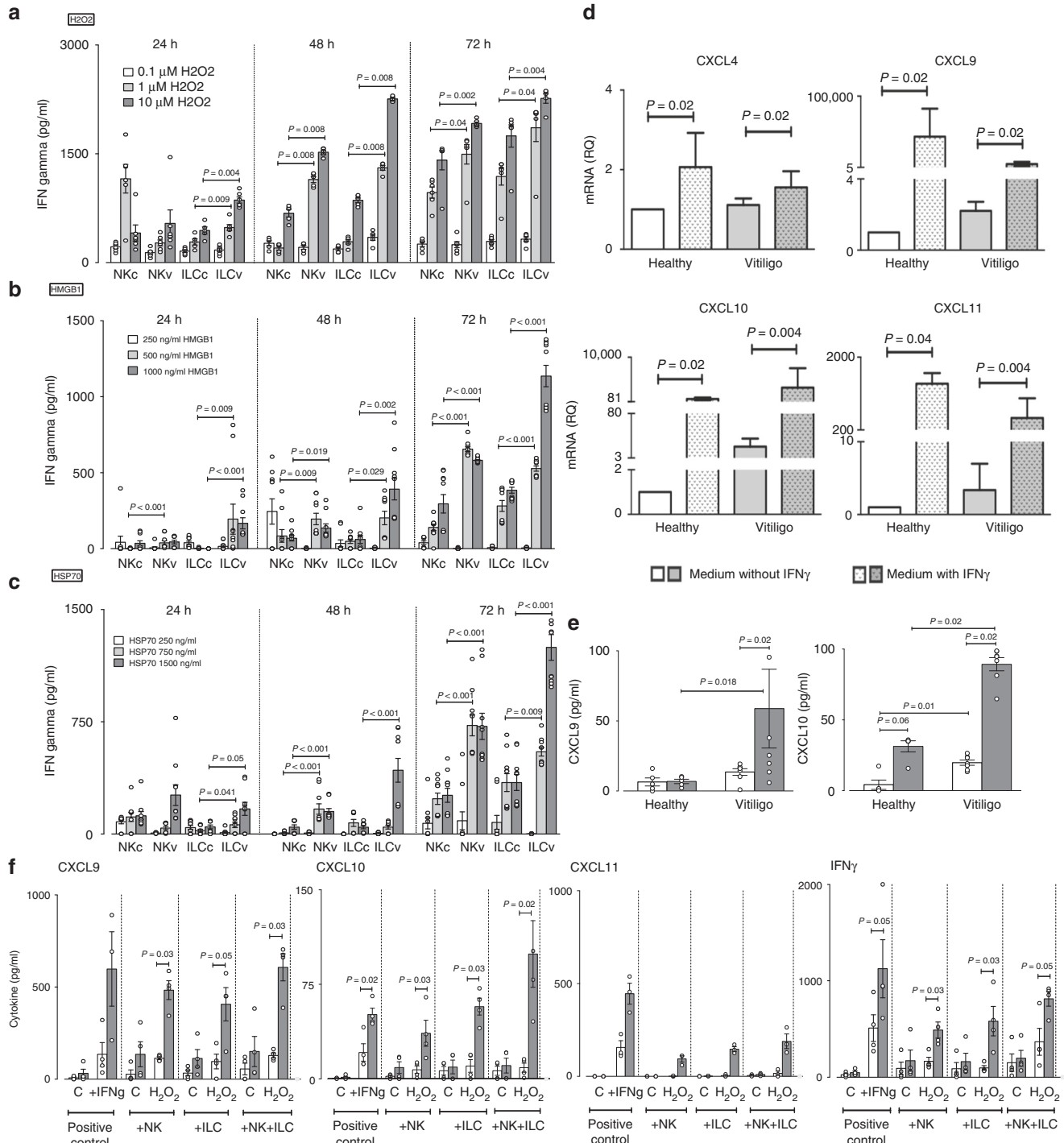

**Fig. 2** Effect of stressed innate immune cells on melanocyte function. IFNγ production by sorted NK and ILCs from blood of healthy control (c, $n = 6$–10) and vitiligo (v, $n = 6$–10) subjects following in vitro stimulation with $H_2O_2$ (0.1–10 μM) (**a**), HMGB1 (250–1000 ng/ml) (**b**) or HSP70 (TKD, 250–1500 ng/ml) (**c**) for 24, 48 or 72 h. The effect of *exogenous* IFNγ (50 ng/ml) on CXCL4, CXCL9, CXCL10, CXCL11 mRNA (**d**) and CXCL9, CXCL10 protein (**e**) production by healthy ($n = 4$–7) and vitiligo melanocytes ($n = 7$–9). In **f** chemokine and IFNγ production was measured 24 h post stimulation of healthy (white bars, $n = 4$) or vitiligo (grey bars, $n = 4$) melanocytes with patient's own *autologous* NK or ILCs (alone or in combination) which were pre-stressed with $H_2O_2$ for 48 h before addition of innate cells to patients own melanocytes. Positive control condition represents melanocytes directly pre-stimulated with IFNγ (50 ng/ml) for the same duration of time. PCR results are normalized to house-keeping gene SB and expressed as fold change in expression relative to the pool of healthy skin samples. Results are shown as individual dot plots with a line either at median (**a**–**c**) or at mean ± SEM (**e**, **f**)

with $H_2O_2$ (to induce oxidative stress in cells). Results have shown $H_2O_2$ to induce IFNγ production in both NK and ILCs and this effect was dose- and time-dependent (Fig. 2a). The stress-induced IFNγ production by NK and ILCs from vitiligo subjects were significantly higher compared to healthy subject's responses at the

same dose (Fig. 2a) suggesting vitiligo innate cells were more sensitive to oxidative stress compared to cells from healthy controls. Innate cells can be activated by many DAMPs. Among them, the role of HSP70 Heat Shock Protein (HSP70) and High Mobility Group Box (HMGB)1 in vitiligo pathogenesis have been suggested

by several studies[27–29]. Thus, we reproduced the experiment with NK and total ILCs from PBMC of healthy (C) and vitiligo (V) patients and this time examined their IFNγ responses after stimulation with HMGB1 (Fig. 2b) or HSP70 (Fig. 2c). Similar to the results obtained following $H_2O_2$ stimulation, HMGB1- and HSP70-induced IFNγ production by NK and ILCs were dose- and time-dependent and responses from vitiligo melanocytes were significantly higher compared to responses in melanocytes from healthy individuals at the same dose.

Next, we set out to examine if primary melanocytes can directly respond to IFNγ. Stimulation of normal human melanocytes (NHM, $n = 5$) with IFNγ for 24 h significantly upregulated their production of CXCL9, CXCL10 and CXCL11 mRNA (Supplementary Fig. 2) ($P < 0.05$). The production of these chemokines was higher in normal human keratinocytes (NHK, $n = 6$), the known potent producers of chemokines, compared to NHM (Supplementary Fig. 2). Knowing that NHM upregulate their chemokine production following IFNγ stimulation, we asked whether this chemokine production varies between healthy and vitiligo melanocytes. Stimulation of healthy or vitiligo melanocytes with IFNγ induced a significant increase in mRNA expression of CXCL4, CXCL9, CXCL10 and CXCL11 (Fig. 2d). However, the production of CXCL9 and CXCL10 was much higher comparatively to CXCL11. Only very limited amount of CXCL4 were detected. Non-stimulated vitiligo melanocytes had significantly higher baseline (or constitutive) production of CXCL10 compared to healthy melanocytes (Fig. 2e). Stimulation with IFNγ significantly increased both CXCL9 and CXCL10 protein in the supernatant of vitiligo melanocytes, producing on average 4–5 fold more chemokines compared to healthy controls (Fig. 2e). Together, these results suggest vitiligo melanocytes are more sensitive to IFNγ stimulation and may have greater chemo-attractive properties.

Next, we wanted to examine if stressed innate immune cells can directly modulate melanocyte function. To do this, we sorted NK and ILC1 from the PBMCs of healthy and vitiligo patients, pre-stressed them in vitro with $H_2O_2$ for 48 h and transferred the innate cells (without media) to their own autologous primary melanocytes. Chemokine production was measured in the supernatant 24 h after co-culture. Results have shown that the addition of pre-stressed innate cells from healthy subjects to their own primary melanocytes did not cause any significant change in melanocyte chemokine production (Fig. 2f). However, addition of pre-stressed NKs or ILCs from vitiligo patients dramatically increased their own melanocyte production of CXCL9, CXCL10, CXCL11 and IFNγ (Fig. 2f). This effect was further increased when both pre-stressed NKs and ILCs were added together and these levels were equal to, or greater than the responses seen when exogenous IFNγ was added to melanocytes (positive control condition). This data suggest that stressed innate immune cells are capable of directly modulating melanocyte function by upregulating their chemokine responses and thereby their chemo-attractive properties. Importantly, these results show that vitiligo melanocytes (compared to healthy melanocytes) are much more sensitive to their own stressed innate immune cells. It is important to note that although the cells were stimulated for 48 h with $H_2O_2$ prior to transfer with melanocytes, these cells were still capable of producing IFNγ and effectively modulating melanocyte function (Fig. 2f). To examine if NKs and/or ILCs are directly capable of producing chemokines in response to stress, we measured the production of CXCL9, CXC10 and CXCL11 by NKs and ILCs after stimulation with HMGB1 or HSP70. NK/ILC production of CXCL9, CXCL10 and CXCL11 following innate stress was negligible (and often undetected in the case for CXCL10) compared to their IFNγ production following the same stress stimuli (Supplementary Fig. 3). Moreover, this NK/ILC

production of chemokines is also negligible compared to the chemokine production by melanocytes (Fig. 2f).

## Human melanocytes express CXCR3B and it's regulated by IFNγ.

CXCR3, a chemokine CXCL9, CXCL10 and CXCL11 receptor, is typically found on T cells, where the predominant isoform expressed is of the CXCR3A form[30]. Whether CXCR3 is expressed on human melanocytes is unknown. Here we demonstrate that melanocytes isolated from healthy human skin express CXCR3, particularly the CXCR3B isoform (Fig. 3). This isoform is absent in mice and therefore not possible to study in animal models of vitiligo. In human skin, CXCR3B was detected at mRNA (Fig. 3a) and protein (Fig. 3b) level in cultured melanocytes and their numbers semi-quantitated in Fig. 3c. We demonstrated melanocytes isolated from vitiligo skin have significantly elevated expression of CXCR3B at baseline compared to healthy control skin (Fig. 3a). IFNγ significantly upregulated CXCR3B mRNA expression in both healthy and vitiligo patients (Fig. 3a). While IFNγ significantly increased the number of CXCR3B + cells in healthy skin, IFNγ had no further effect on vitiligo melanocytes whose CXCR3B expression was already high (Fig. 3c). Expression of CXCR3B in healthy human keratinocytes was significantly lower than the expression in healthy melanocytes which was confirmed at both mRNA and protein level (Fig. 3a, b). Interestingly, IFNγ had no effect on keratinocyte expression of CXCR3B (Fig. 3a, b). Finally, we have demonstrated that there is an increased number of MITF + CXCR3B+ melanocytes in the NL skin of vitiligo patients compared to healthy skin ($P = 0.03$) (Fig. 3d). MITF (Microphthalmia-associated transcription factor), is a specific melanocyte marker and a master regulator of melanocyte development.

## CXCL10 activates melanocytic CXCR3B to induce apoptosis.

From previously published literature in human breast cancer[31] there is evidence to suggest that CXCR3B-mediated signal is pro-apoptotic. To examine the function of CXCR3 on viability of human vitiligo melanocytes, real-time detection of melanocyte death was monitored before and after exposure to CXCL10 using IncuCyte® live cell imaging system. Stimulation with CXCL10 caused a significant dose-dependent increase in melanocyte death compared to non-stimulated melanocytes but only in IFNγ primed melanocytes ($P = 0.006$, Fig. 4a). As CXCR3B-specific antagonists do not exist yet, initially we tested the effect of a broad CXCR3 antagonist AS612568 on CXCL10-induced melanocyte death. We have demonstrated AS612568 to inhibit melanocyte death induced by CXCL10 in a dose-dependent manner, completely abolishing CXCL10-induced effects at 2 μM ($P = 0.045$, Fig. 4a). Following these positive results, we set out to use a more targeted approach transfecting the melanocytes with siCXCR3 (or siControl). The knockdown efficiency of our siCXCR3 was determined at both mRNA (Supplementary Fig. 4a) and protein (Supplementary Fig. 4b) level, achieving >80% silencing of the CXCR3B gene with 80 nM siCXCR3. Our results have shown that in IFNγ pre-treated, siControl transfected cells, CXCL10 significantly increased the death of melanocytes (demonstrated by increased number of yellow-immunoreactive cells in IncuCyte, Fig. 4b). This death was seen in melanocytes extracted from both healthy ($P = 0.008$) and NL vitiligo skin ($P = 0.03$) (Fig. 4c). Vitiligo melanocytes were significantly more sensitive to CXCL10-induced death compared to healthy melanocytes ($P = 0.004$), inducing ~2-fold difference in rate of melanocyte death (Fig. 4c). Transfection of melanocytes with siCXCR3 prior to CXCL10 stimulation, completely inhibited the CXCL10-induced death and restored baseline responses in both healthy ($P = 0.004$) and vitiligo ($P = 0.03$) melanocytes (also

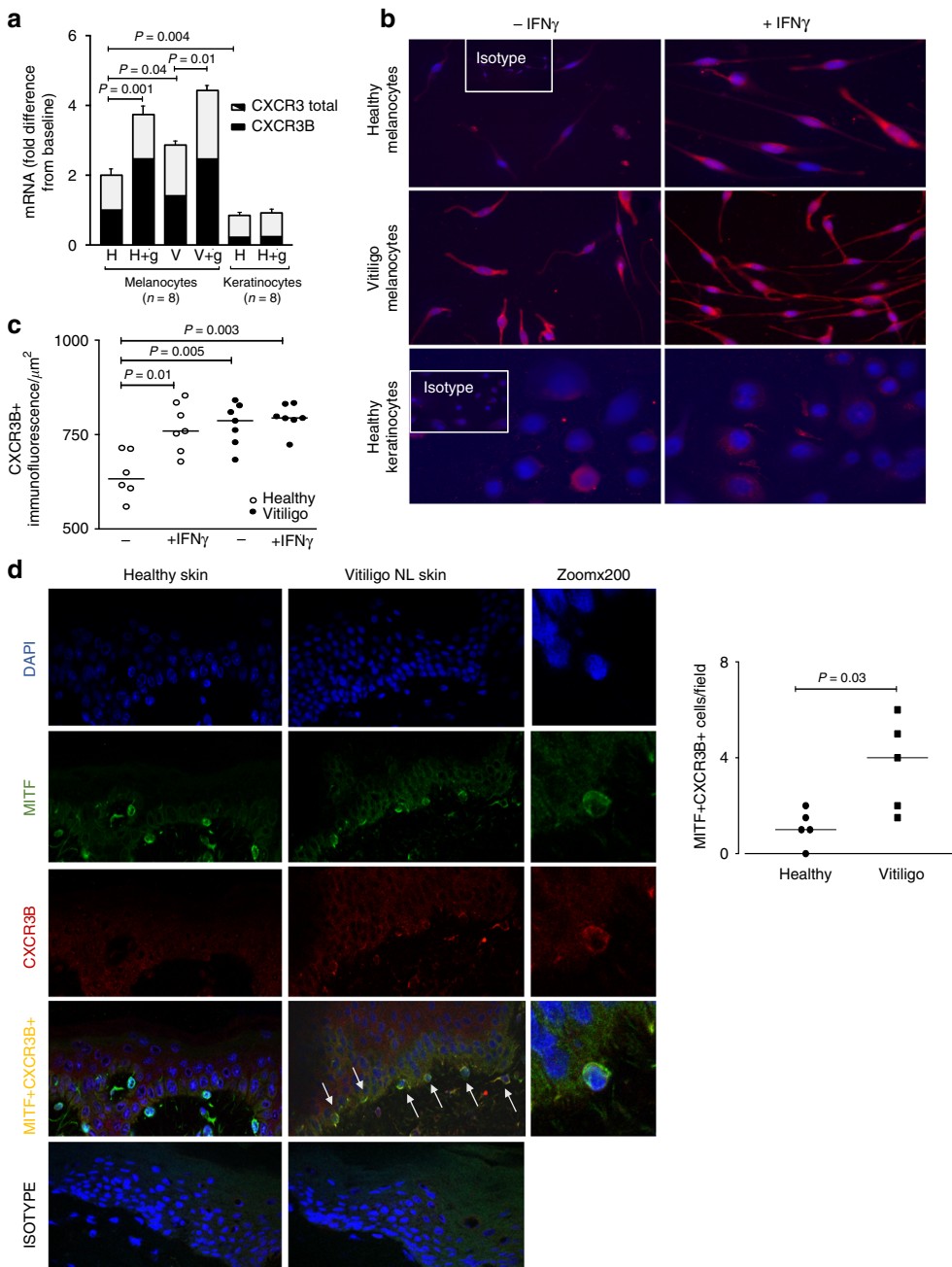

**Fig. 3** Expression of CXCR3B in human melanocytes and their regulation by IFNγ. **a** Total CXCR3 mRNA (black + white bars) and CXCR3B mRNA (black bars) in healthy and vitiligo primary melanocytes ($n = 8$) and healthy keratinocytes ($n = 5$) before and after exposure to IFNγ 50 ng/ml for 24 h. Results are normalized to unstimulated melanocytes from healthy subjects and expressed as mean ± SEM. **b** Immuno-detection of CXCR3B protein and **c** its semi-quantification in melanocytes extracted from healthy skin (open circles, $n = 6–7$) and skin from vitiligo patients (closed circles, $n = 7$) before and after IFNγ stimulation. **d** In situ detection of CXCR3B+ melanocytes (MITF+CXCR3B+) in vitiligo non-lesional (NL) skin and their quantification ($n = 5$) compared to healthy skin ($n = 5$). Results are shown as individual dot plots with a line at median

visualised in Fig. 4b by reduction in yellow-stained dead cells). To more specifically examine the contributing role of CXCR3B, we repeated this experiment in both healthy and vitiligo melanocytes using custom design specific silencer RNA directed against the CXCR3B isoform (efficiency of the SiCXCR3B at protein level shown in Supplementary Fig. 4c). Transfection of melanocytes with siCXCR3B prior to CXCL10 stimulation, significantly reduced the CXCL10-induced death in both healthy ($P = 0.002$) and vitiligo ($P = 0.001$) melanocytes (Fig. 4d). In addition to CXCL10, we examined responses to CXCL9 and CXCL11. Treatment with CXCL9 and CXCL11 also induced a significant

melanocyte death in both healthy ($P = 0.034$ and $P = 0.007$, respectively) and vitiligo ($P = P < 0.001$ and $P = 0.01$) melanocytes that was almost completely prevented by the use of SiCXCR3B (Fig. 4d). Interestingly, CXCL9-induced melanocyte death was significantly lower compared with CXCL10-induced death in both healthy ($P < 0.001$) and vitiligo ($P = 0.0035$) patients. The difference was even more pronounced with CXCL11, that induced much lower cell death compared to CXCL10 ($P < 0.001$ for both healthy and vitiligo).

It has been shown that the activation of CXCR3B in breast cancer cells activates p38 MAPK and PARP pathway, inducing

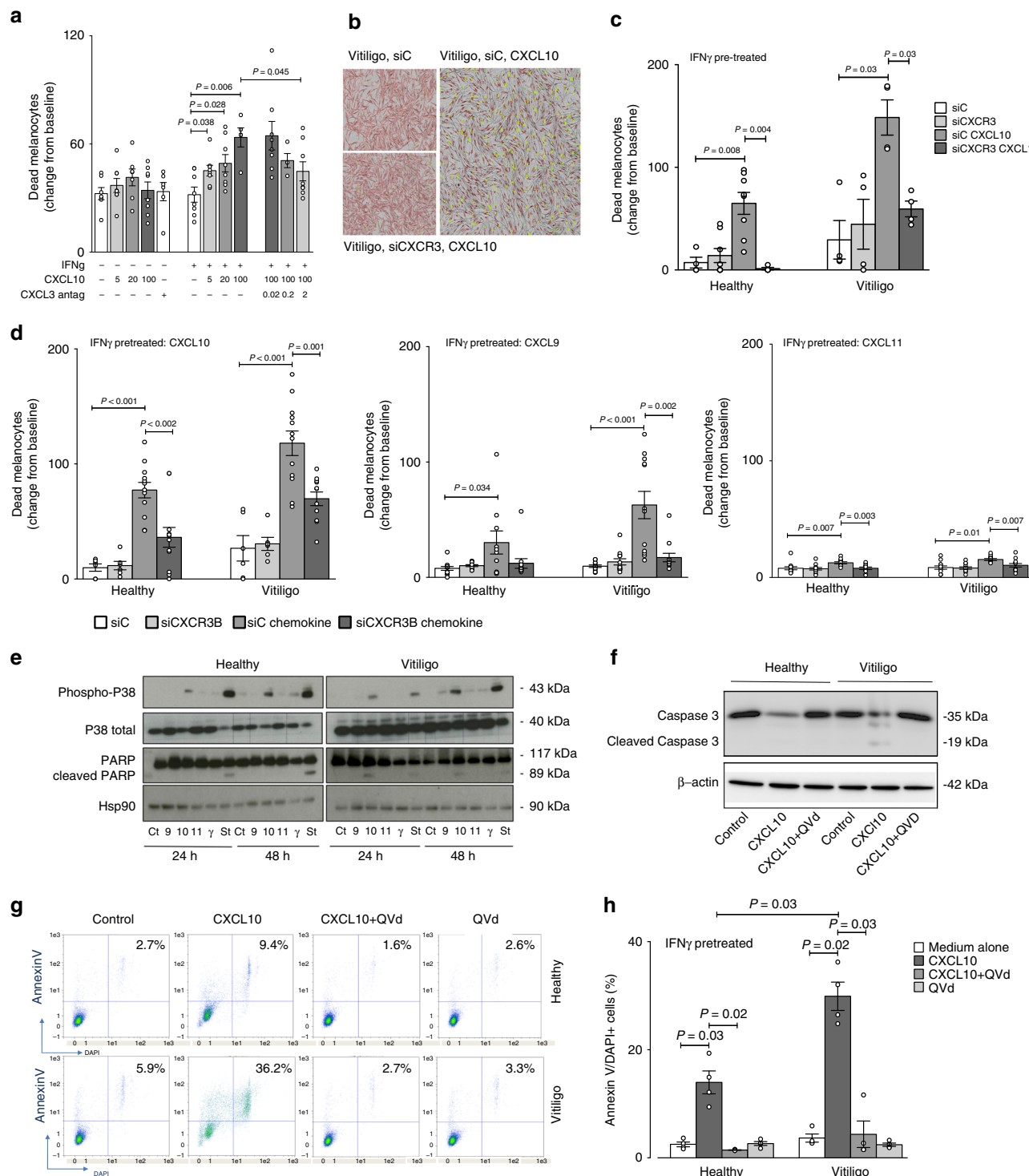

cell apoptosis[31]. Studying the signalling pathway involved in chemokine-induced CXCR3B activation in melanocytes, we have shown that in melanocytes extracted from healthy patients and not pre-treated with IFNγ, only a slight activation of p38 but not PARP cleavage was seen following chemokine stimulation (Fig. 4e). This was in contrast to vitiligo patients where we observed increased activation of p38 and PARP cleavage. In accordance to the melanocyte cell death observed using IncuCyte experiment, the activation of the pathway is mostly observed after CXCL10 treatment compared to CXCL9 or CXCL11 stimulation (Fig. 4e). CXCL10-induced changes were of the similar magnitude as those seen following stimulation with positive

control staurosporine (St), a potent pro-apoptotic therapeutic agent (Fig. 4e). Interestingly, the absence of CXCL10-induced apoptosis in NHM without IFNγ pre-treatment (Fig. 4e) while a significant apoptosis is observed when they are pre-treated with IFNγ (Fig. 4c), is in accordance with the low basal expression of CXCR3B in NHM and its increased expression after IFNγ stimulation (Fig. 3b, c). To assess the role of apoptosis in CXCL10-induced melanocyte death, in parallel we have shown cleavage of caspase-3 in vitiligo patients but not in healthy controls and this effect to be reversed by QVd-OPh, a caspase inhibitor (Fig. 4f). To further understand this CXCL10-induced death mechanism, we pre-treated healthy and vitiligo

**Fig. 4** CXCL10 activates melanocytic CXCR3B to induce apoptosis. **a** Effect of CXCL10 (5, 20, 100 pg/ml) on cell viability in unstimulated or IFNγ stimulated (50 ng/ml for 48 h) melanocytes from vitiligo patients, in presence or absence of CXCR3 antagonist AS612568 (0.02 μM, 0.2 μM or 2 μM) ($n = 5$–8). Cell viability was monitored using IncuCyte® live cell fluorescence imaging system. **b** Illustrates live IncuCyte images of vitiligo melanocytes 24 h after exposure to CXCL10 in cells transfected with siC or siCXCR3. Melanocytes were tracked with CellTracker™ Red CMPTX dye and dead cells tracked with IncuCyte® Cytotox Green reagent. Co-localised yellow cells represent dead melanocytes. The effect of siCXCR3 (or its siC) on CXCL10 (100 pg/ml)-induced death of healthy ($n = 4$–8) and vitiligo ($n = 4$) melanocytes are shown in **c**. In **d** healthy and vitiligo melanocytes ($n = 6$–12) were transfected with siCXCR3B (or its siC) and melanocyte death shown at 24 h following CXCL10, CXCL9 or CXCL11 (100 pg/ml) stimulation. In separate experiments, cell lysates was used to study the signalling pathway induced by chemokines, IFNγ (50 ng/ml) or Staurosporine (positive control, 1 μg/ml) at 24 or 48 h post stimulation, measuring the expression levels of phosphorylated and total p38 and total and cleaved poly(ADP-ribose) polymerase (PARP) by Western Blot analysis (**e**). HSP90 was used as an internal loading control. Representative blot of 3 separate experiments is shown. Total and cleaved caspase-3 activity (**f**) and proportion of apoptotic cells (counted as % Annexin V+DAPI+cells in FACS analysis) (**g**) from healthy ($n = 4$) and vitiligo ($n = 4$) melanocytes stimulated with 100 pg/ml CXCL10 for 24 h in presence or absence of QVd OPh (10 μM, caspase inhibitor) (**h**). Results are shown as individual dot plots with a line at mean ± SEM

melanocytes with IFNγ prior to CXCL10 and QVd stimulation, and using FACS, we have demonstrated CXCL10 to significantly increase the percentage of AnnexinV + DAPI+ apoptotic cells in both healthy (2.7–9.4%) and vitiligo melanocytes (5.9–36.2%) (Fig. 4g). The level of apoptosis was>3-fold greater in vitiligo ($P = 0.02$) compared to healthy cells ($P = 0.03$) and apoptosis was completely reversed by QVd in both groups ($P = 0.03$, Fig. 4g, h). IFNγ or QVd alone or CXCL10 in non IFNγ primed cells had no effect on apoptosis. It is interesting to note that as a comparison, we detected low expression of CXCR3B on CD4+ and CD8+ lymphocytes isolated from PMBCs of healthy patients (Supplementary Fig. 5a), however while CXCL10 stimulation of T cells had no effect on T cell death, CXCL10 stimulated proliferation of both CD4 and CD8 T cells (Supplementary Fig. 5b).

**CXCR3B-induced melanocyte death triggers adaptive immunity**. To examine whether CXCR3B-induced melanocyte death following pre-treatment with IFNγ and stimulation with CXCL10 may initiate adaptive melanocyte auto-immunity by their own presentation of antigens, we replenished media post CXCL10 stimulation and 3 days later stimulated melanocytes with their *autologous* T cells. Our IncuCyte® results have shown that there was significantly higher melanocyte death when T cells were present with CXCL10-stimulated melanocytes compared to melanocyte death seen with CXCL10 stimulation alone ($P = 0.02$) or addition of T cells alone ($P = 0.02$) (Fig. 5a). These results were reproducible and were of the similar magnitude as seen following stimulation of melanocytes with *allogenic* T cells (Supplementary Fig. 6). Interestingly, pre-incubation of T cells with CXCL10 for 24 h, prior to their addition to IFNγ-primed melanocytes did not induce melanocyte death while the same T cells added to IFNγ-primed melanocytes treated with CXCL10 did suggesting lack of direct effect of CXCL10 on the T cells (Supplementary Fig. 7). T cell enhanced melanocyte death in IFNγ-primed melanocytes was accompanied by increased melanocyte expression of co-stimulatory (CD40, CD80, HLA-DR) and adhesion (ICAM-1) molecules (Fig. 5b) and parallel increase in absolute number of CD3+ T cells in IncuCyte co-cultures with time ($P = 0.03$) (Fig. 5c). These CD3+ T cells were indeed shown to be Ki67+ proliferating cells (Fig. 5d). These data are supported by flow cytometric quantification of T cell proliferation which showed an increased number of dividing T cells (evaluated as %CD3 + CFSE+ cells) when they were co-cultured with IFNγ-primed NHM stimulated with CXCL10 and responses were similar to that seen following PHA stimulation (Fig. 5e). This degree of proliferation was not seen when T cells were co-cultured with non-primed NHM exposed to CXCL10. Treatment with CXCR3 antagonist prevented the potentiating effect of T cells on CXCL10-induced melanocyte death ($P = 0.008$) (Fig. 5a), augmentation of T cell number ($P = 0.04$) (Fig. 5c) and T cell proliferation (Fig. 5d, e).

## Discussion

Here we confirm the previously reported increased level of NK cells in the blood of vitiligo patients[16], and extend it to ILC1. Our transcriptome data further suggest that NK and ILC1 cells are also increased in NL skin of vitiligo patients. Interestingly, this subtype is associated with the Th1 response which is known to be the key adaptive pathway in vitiligo[7,13]. When these cells are cultured under stress conditions, they produce IFNγ which induces CXCLs production. Here we show that NK and ILCs from vitiligo patients produce higher amount of IFNγ under oxidative stress but also under the stimulation of the two main DAMPs implicated in vitiligo pathophysiology, namely HSP70 and HMGB1 compared to healthy subjects. Thus, not only vitiligo patients have an increased population of NK and ILC1 in their skin and their blood, but these cells are more sensitive to DAMPs compared to healthy individuals.

It has been previously shown that skin memory T regulatory cells express CXCR3, that these cells are expressed in vitiligo skin during the depigmenting process and they remain in the skin even after repigmentation[32]. Those cells can be an important source of IFNγ production and their presence in repigmented areas could explain the recurrences of vitiligo lesions which often affected the same skin locations. However, skin memory T regulatory cells need that the adaptive immunity occurs to be recruited and thus can't be the initial source of IFNγ production. Thus, they can't explain the initial steps that lead to the recruitment and activation of T cells against melanocytes and the development of the first clinical lesions. NK and ILC1 cells are potent producer of IFNγ under endogeneous or exogeneous stress conditions. Their significant increase in the blood and in the NL vitiligo skin argues for their key role in the initial production of IFNγ. Importantly, we showed that melanocytes express CXCR3, more particularly the CXCR3B isoform. Also known as G protein-coupled receptor 9 (GPR9), CD183, P-10 receptor, and MIG receptor, CXCR3 is a chemokine receptor expressed on autoreactive T cells that have been implicated in a range of physiological processes and related disorders. CXCR3 is largely absent from naive T cells but is upregulated upon activation with antigen and recruits activated cells to sites of tissue inflammation in response to its primary ligands: CXCL9, CXCL10, and CXCL11. CXCL9 and CXCL10 are strongly produced in many autoimmune and inflammatory diseases leading to the accumulation of CXCR3+T cells with Th1 phenotypes[33]. CXCR3 has three isoforms in human: CXCR3A, CXCR3B and CXCR3Alt[34]. Importantly, the isoform B is absent in rodents therefore the expression nor the role of CXCR3B could be investigated in vitiligo animal models, thus requiring to be studying in human samples. The isoform CXCR3A is expressed by T lymphocytes and has an important role in the adaptive immune system[30]. CXCR3A activation induces differentiation and proliferation.

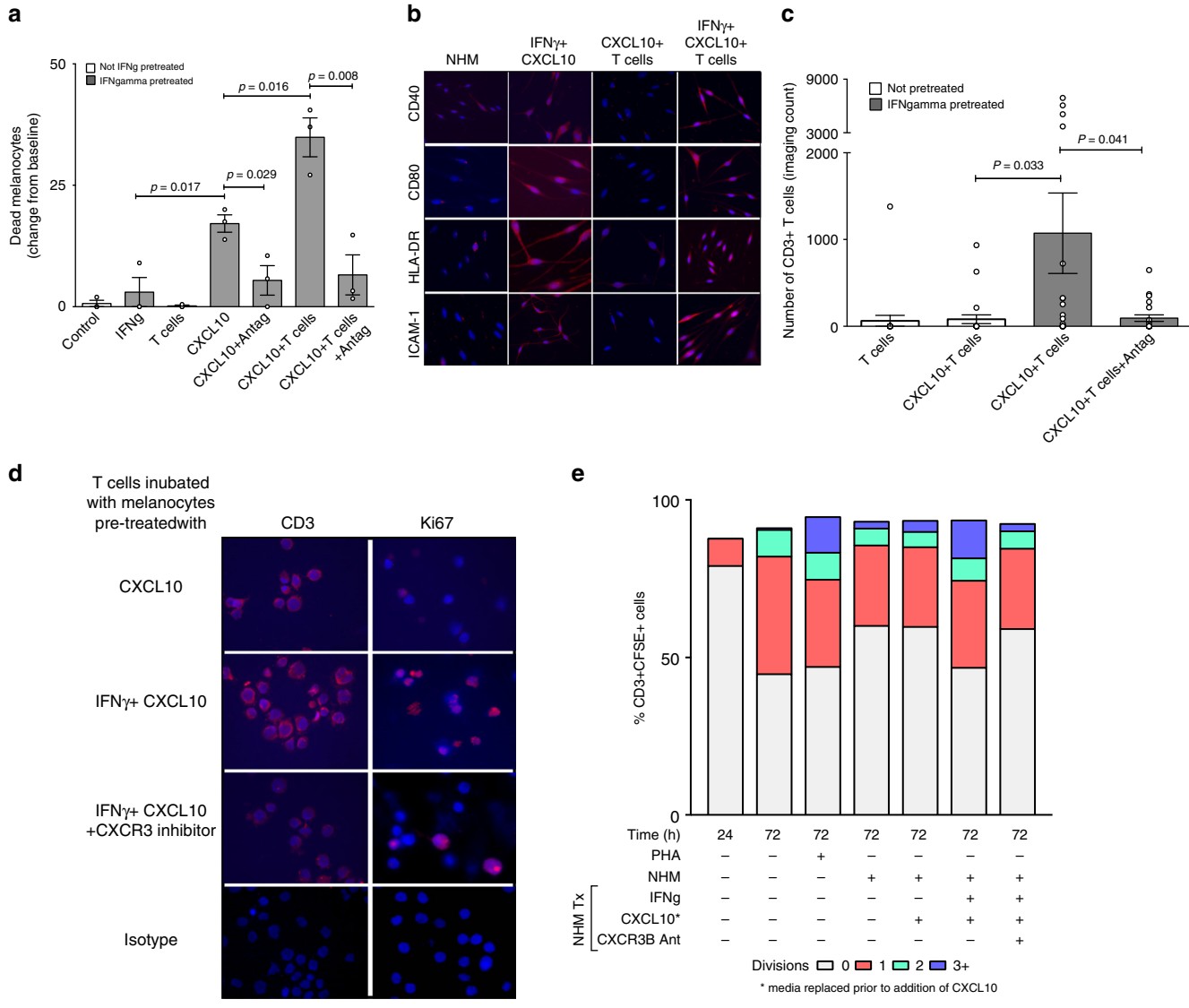

**Fig. 5** T cells enhance CXCL10-induced melanocyte death by induction of adaptive immunity. **a** CXCL10-induced death of vitiligo melanocytes in presence or absence of patients own *autologous* T cells. As above, IFNγ-pretreated melanocytes were exposed to CXCL10 in presence or absence of CXCR3 antagonist AS612568 (2 μM). The next day, media was replaced and 3 days later patient's own CD3+ T cell were sorted and added to the melanocytes (n = 3) prior to initiation of IncuCyte. **b** At the end of the experiment (~48 h), supernatant was collected, remaining cells trypsinised and cytospin sections prepared for immunofluorescence detection of co-stimulatory (CD40, CD80, HLA-DR) and adhesion (ICAM-1) molecules on melanocytes. **c** T cell-induced potentiation of melanocyte death in IFNγ-pretreated melanocytes (compared to untreated melanocytes) was associated with a parallel increase in the number of CD3+ T cells which was supported by increased expression of Ki67+cells in the same cytospin sections **d**. Results are shown as individual dot plots with a line at mean ± SEM. **e** T cell proliferation was quantified by flow analysis measuring the percentage of CD3+CSFE+ cells undergoing 0, 1, 2 or 3+divisions (n = 3). Labelled cells at time zero was used as a negative reference, unstimulated cells left in culture for 72 h before labelling as a control and cells stimulated with PHA for 72 h (Phytohemaglutinin, 5 μg/ml) as a positive control

CXCR3B is only marginally expressed at the surface of immune cells and its activation induces apoptosis. We show here that CXCR3B expression is low at baseline in melanocytes extracted from healthy subjects and increases under IFNγ stimulation. Melanocytes from vitiligo patients have elevated basal expression of CXCR3B. Interestingly, we also observed a significant increase in CXCR3B expression in melanocytes from NL skin of vitiligo patients compared to the skin of healthy volunteers, reinforcing the data showing that melanocytes from vitiligo patients have intrinsic defects[19]. We demonstrate that CXCL10 activates the downstream pathway of CXCR3B in melanocytes of vitiligo patients, leading to their apoptosis. CXCL9, and in a much lesser extent CXCL11, are also able to induce some CXCR3B-induced apoptosis of melanocytes, but in a much less effective manner

compared to CXCL10. It has previously been demonstrated that CXCL10 has a much greater binding affinity for CXCR3B compared to CXCL9 or CXCL11 while the affinity is similar between the three chemokines for the CXCR3A isoform[35]. This may be attributed to differences in special conformation between CXCR3A and CXCR3B[36].

We show that CXCL10-induced apoptosis occurs to a greater extent in melanocytes extracted from vitiligo patients and that this apoptosis can be prevented by using CXCR3 antagonists, by silencing CXCR3B, or by using caspase-inhibitors. Few data are available on the expression and the role of CXCR3B in non-immune cells, however CXCL10 was shown to inhibit angiogenesis by activating CXCR3B at the surface of endothelial cells and inducing their apoptosis[37]. We further demonstrate that

under IFNγ and CXCL10 stimulation, several vitiligo melanocytes undergo apoptosis when the remaining ones become activated expressing CD40, CD80, HLA-DR (MHC-II) and ICAM-1 at their surface and present their own antigens, subsequently triggering T cell proliferation and the onset of the adaptive response by the lymphocytes. The fact that melanocytes may become antigen presenting cells upon IFNγ stimulation was suggested years ago[21]. We show here that not only vitiligo melanocytes undergo initial apoptosis through the CXCR3B activation at their surface but may also be the presenting cells that initiate the adaptive response against themselves. This CXCR3B-induced apoptosis that occurs before T cells are attracted could also explain the reduced number of melanocytes that is observed in NL vitiligo skin[38]. Thus, both phenotypically normal melanocytes as well as NK and ILC cells from vitiligo patients respond differently to various stimuli when compared to cells from healthy donors. These data suggest that all these cells harbour genetic abnormalities. These results are in accordance with the genetic studies which reported allelic variations in innate and adaptive immune genes but also in melanocytic genes[39].

Taken together, our results shed light on the initial events that trigger the anti-melanocyte auto-immunity and give light to the melanocytes as a central player in the initiation of this process. Upon endogenous or exogenous stress, NK and ILC1 produce IFNγ that induce the production of chemokines by not only keratinocytes but by melanocytes. These chemokines attract and activate T cells and furthermore, CXCL10 induces the initial apoptosis of some melanocytes resulting in release of auto-antigens. Activated by the local production of IFNγ, the remaining melanocytes express co-stimulatory molecules (CD40, CD80, MHC-II, ICAM-1) and present their own antigens to the T cells attracted by the chemokines and initiate the adaptive response (Fig. 6). These results emphasize the key role of CXCR3B in the initial apoptosis of melanocytes and identify this receptor as a target to prevent and to treat vitiligo by acting at the very early steps in the destruction of melanocytes.

## Methods

**Biological samples and experimental outline**. A total of 21 subjects were recruited for this study (Table 1). Fourteen vitiligo patients ($n = 14$) with stable disease were enrolled (Department of Dermatology, L'Archet Hospital, CHU Nice) after informed, written consent was obtained. Eight healthy controls ($n = 8$) were recruited from the same clinic who were surveilled for skin cancer (suspicious biopsy or familiar history) but were negative on examination. The two groups were matched for gender, age, and location. The study was approved by the National ethics committee (N12.034). From each vitiligo patient, we obtained $3 \times 4$-mm skin punch biopsies (1 lesional, L and 2 non-lesional, NL) and for each healthy control 2 NL biopsies. Fifty (50) ml of blood was obtained from all subjects. L and NL skin was processed for immunohistochemistry and the remaining NL biopsy used to extract primary melanocyte from the skin which were then propagated in vitro. Blood was used to isolate peripheral blood mononuclear cells (PBMCs) by Ficoll gradient centrifugation (Lymphoprep®, Euromedex, France). Ten million PBMCs were used immediately for FACS analysis to examine the phenotype and frequency of NK/ILC in the blood (for markers see below). The remainder of PBMCs ($30-40 \times 10^6$) were frozen down in FCS 10% DMSO and kept at $-156\,°C$ for subsequent sorting of NK/ILC populations and co-culturing with patients own *autologous* primary melanocytes extracted from their skin biopsies. Propagation of primary melanocytes in vitro takes 2–3 months to obtain sufficient number of melanocytes required for the experiment.

**Immunofluorescence and immunohistochemistry**. One L and NL skin biopsy from each patient was embedded in Tissue-Tek® O.C.T medium in cryomolds (Sakura Finetek, USA), snap-frozen in liquid nitrogen-cooled isopentane (2-methyl butane) and stored at $-80\,°C$ before cutting. Tissue was cut with microtome into 7 μm sections and fixed on Superfrost Plus slides. After 20 min of air-drying, sections were fixed in ice-cold acetone (30 min at $-20\,°C$) and saturated for 2 h at RT with PBS 5% BSA 3% normal goat serum. For detection of Granzyme B+ NK cells and IFNγ-producing NK cells, we incubated sections overnight at 4 °C with specific primary antibodies directed against human CD3, CD56, Granzyme B or IFNγ (Abcam and all used at 1:100). The next day the signal was revealed by incubating for 1 h at RT with secondary antibodies conjugated to

fluorochromogens (1:1000) and slides mounted with Prolong® Gold Antifade Reagent with DAPI (Thermofisher). CD3-CD56+ NK cells were visualized as red immunostained cells, Granzyme B and IFNγ as green and yellow as co-localised NK cells containing Granzyme B or IFNγ. In isotype controls, the primary antibody was omitted. To confirm immunofluorescence results, skin slides were stained with immunohistochemical markers directed against human CD3, CD56 and Granzyme B using automated staining protocols (Dako Autostainer Link 47) and results were interpreted by pathologist Dr. Nathalie Cardot-Leccia (Pasteur Hospital, CHU Nice). In situ ILC1 population was detected using anti-human CD161 (green) (Novus Biological, US) and anti-Tbet (BioLegend, London, UK) (red) with the use of True-Nuclear® Transcription Factor Buffer Set (Biolegend) for nuclear staining. CD161 + Tbet+ cells were visualized as red and confirmed to be CD3, CD19 and CD14 negative but CD127+. Semi-quantitative analysis of all immunofluorescence images and positive immunoreactive cells were counted over 10 non-overlapping fields in 5–8 patients/group using either Nikon A1R confocal microscope or Leica microscope, both at 40× objective.

**FACS analysis of NK and ILC populations in the blood**. Total NKs were defined as CD3-CD56+ cells. Cytotoxic or cytokine-producing NKs were differentiated based on their additional CD16 expression whereby CD56$^{bright}$ CD16$^{dim}$ cells are cytokine producing NKs and CD56$^{dim}$ CD16$^{bright}$ are cytotoxic NK cells. Gating strategy for ILC subpopulations initially involved gating all live PBMCs for negative Lineage selection (to exclude CD3+ T cells, CD19+ B cells, CD14+ macrophages, CD34+ eosinophil progenitors, CD123+ dendritic cells and TCR). Lin- cells were then selected for CD127 positivity which stains for all ILCs but not NK cells which are CD127-. Finally, CD117 (c-kit) and CRTh2 (prostaglandin D2 receptor, CD294) markers were used to delineate between the 3 ILC subclasses whereby ILC1 were defined as Lin-CD127 + CRTh2-CD117-, ILC2 as Lin-CD127 + CRTh2 + CD117+ and ILC3 as Lin-CD127 + CRTh2-CD117+ cells. All antibodies were purchased from Miltenyi (Paris, France). Gating strategies are shown in Supplementary Fig. 1.

**Isolation of primary human melanocytes**. On arrival, skin biopsies were rinsed in 70% ethanol followed by $2 \times$ PBS 1% Antibiotic-Antimycotic solution (Gibco), prior to dissociation of dermal-epidermal junctions by overnight digestion in a Dispase solution (Life Technologies) at 4 °C. The next day, dermis was discarded and epidermis digested in a trypsin/EDTA solution for 20 min at 37 °C. Cellular suspension was passed through a 70 μm filter and melanocytes re-suspended in MCDB 153 medium (Sigma-Aldrich, St. Louis, MO, USA) supplemented with 2% fetal bovine serum (FBS; Perbio, Cramlington, UK), 5 μg/ml insulin (Sigma-Aldrich), 0.5 μg/ml hydrocortisone (Sigma-Aldrich), 16 nM tetradecanoylphorbol-13-acetate (TPA) (Sigma-Aldrich), 1 ng/ml fibroblast growth factor (FGF; Promega, Madison, WI, USA), 15 μg/ml bovine pituitary extract (Invitrogen, Waltham, MA, USA) and 10 μM forskolin (Sigma-Aldrich). Melanocytes were maintained at 37 °C in 95% O$_2$/5% CO$_2$ atmosphere and supplemented with 0.08% G418 geneticin (20 μg/ml, Invitrogen) for ~2 weeks to eliminate rapidly growing cells (i.e., keratinocytes hence selecting for melanocyte propagation). Once selected, melanocytes were grown in Cascade Biologics 254 medium (ThermoFisher Scientific) with Human Melanocyte Growth Supplement (HMGS, Gibco).

**Response of NK and ILCs to stress**. NK and total ILC were sorted (SH800S Cell Sorter, Sony Biotechnology) from PBMC of healthy and vitiligo patients using the same strategy as for FACS analysis (Supplementary Fig. 1). Sorted NKs were plated in round-bottom 96-well plates in RPMI 10% FCS prior to stimulation with three different stress-inducing DAMP molecules including H$_2$O$_2$ (0.1, 1, 10 μM) (Sigma-Aldrich Chimie, Lyon France), HMGB1 (250, 500 or 1000 ng/ml) or HSP70 (TKD peptide at 250, 750 or 1500 ng/ml) (both from Abcam, Cambridge, UK). Supernatant was collected at 24, 48 or 72 h post stimulation. IFNγ and chemokine (CXCL9, CXCL10 and CXCL11) production was measured using commercially available ELISA kits (PeproTech, NJ, USA).

**Response of melanocytes and keratinocytes to exogenous IFNγ**. NHM and NHK were isolated from neonatal foreskin obtained after circumcision as previously described. Cells were stimulated with IFNγ (50 ng/ml, Miltenyi Biotec) for 24 h. Cell pellets were harvested for extraction of RNA (using RNeasy kits, Qiagen) and 1 μg RNA used to synthesize cDNA using the Reverse Transcription System (Promega) which was then used as a template for amplification by real-time qPCR with Sybr$^{TM}$ Green reagent (Life Technologies, CA, USA) and specific primers directed against CXCL4, CXCL9, CXCL10 and CXCL11 genes. All measurements were performed in triplicate and results normalized to the expression of the SB34 housekeeping gene. In separate experiments, primary melanocytes from healthy and vitiligo skin biopsies were also stimulated with IFNγ and 24 h later both cell pellets and supernatants were collected for measurement of chemokine production at mRNA and protein level.

**ILC and melanocyte co-culture**. The supernatant from the NKs and total ILCs which were previously sorted from PBMCs and stressed with 1 μM H$_2$O$_2$ for 48 h was replaced with fresh media prior to cells being directly added to their own primary *autologous* healthy or vitiligo melanocytes. Melanocytes were also

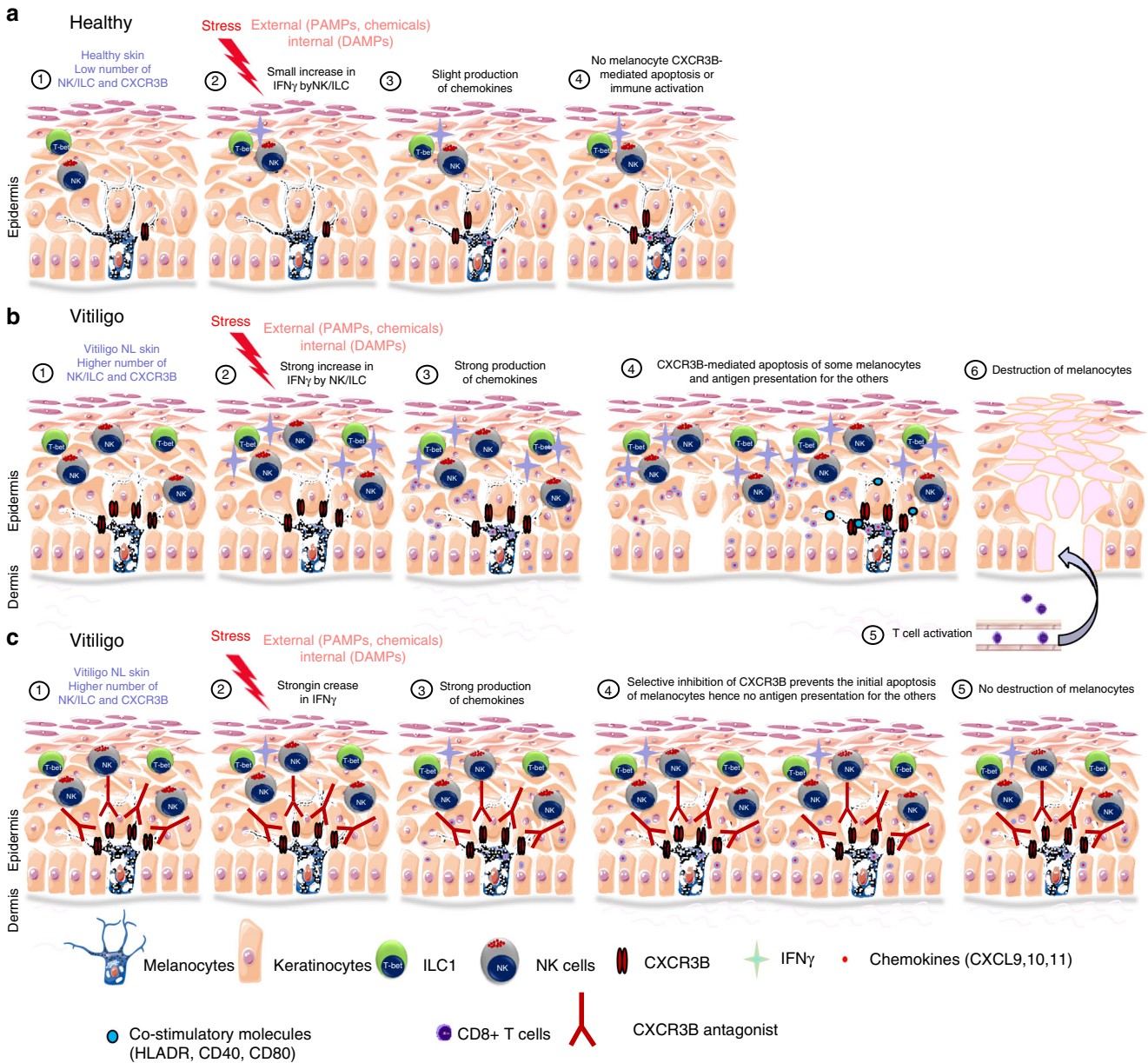

**Fig. 6** Initial steps in the anti-melanocytic immune activation in vitiligo. **a** Schematic representation of normal skin. A low number of NK and ILC1 cells are present in the skin. External stress on the skin induces only a small increase in IFNγ by NK/ILC which is not sufficient to initiate their chemokines production, hence not initiating melanocyte apoptosis or immune activation. **b** Schematic representation of vitiligo skin. The basal number of NK and ILC1 cells is increased. Damage-associated molecular pattern (DAMP) molecules of genetically predisposed melanocytes can activate ILC1 and NK (1). Alternatively, ILC1 and NK can be also activated by external stress and pathogen-associated molecular pattern molecules (PAMP) (2). These NK and ILC cells are more sensitive to stress compared to healthy individuals and in response, they produce large amount of IFNγ (2) that induces the secretion of chemokines by keratinocytes and to a lesser extent by melanocytes (3). These chemokines (mostly CXCL10), bind chemokine receptor CXCR3B at the surface of melanocytes and induces their initial apoptosis and release of melanocytic antigens. The remaining melanocytes stimulated by IFNγ express co-stimulatory (CD40, CD80, HLA-DR) and adhesion (ICAM-1) molecules and start presenting their own antigens to the naive T cells attracted by the chemokines (5), thus initiating the initial steps involved in anti-melanocytic immunity in the skin (6). **c** Schematic representation of vitiligo skin treated with a selective inhibition of the B isoform of CXCR3. DAMPs and PAMPs induce a strong production of IFNγ by NK and ILC1 cells followed by a potent production of chemokines. However, the selective inhibition of CXCR3B prevents the initial melanocytic apoptosis and subsequent presentation of melanocytic antigens without impairing T cell function

incubated with *autologous* non-stressed NKs or total ILCs as controls. Positive controls were wells directly stimulated ex vivo with 50 ng/ml IFNγ. Supernatant was collected 24 h post incubation with innate cells or IFNγ to measure melanocyte production of CXCL9, CXCL10, CXCL11 and IFNγ by ELISA.

**Detection of CXCR3B on human melanocytes.** Melanocytes from healthy and vitiligo subjects were stimulated with 50 ng/ml IFNγ for 24 h and cell pellets harvested for RNA extraction. Real-time qPCR was performed using specific primers directed against CXCR3 total or against CXCR3B and results normalized to the house-keeping gene SB 34 (Supplementary Table 1). CXCR3B mRNA levels were compared to expression in healthy human keratinocytes at baseline and following IFNγ stimulation. In separate experiments, melanocytes and keratinocytes were grown on cover slides in 12-well plates and stimulated with IFNγ as above. Twenty-four hours later, they were fixed with 1% paraformaldehyde, saturated in PBS 3% BSA containing 3% goat serum for 1 h and incubated overnight at 4 °C with primary antibody directed against CXCR3B (1:200) (Proteintech, USA). The next day, slides were washed 3× with PBS 3% BSA 0.1% Tween20 prior

to incubation for 1 h with secondary antibody (goat anti mouse AF594, 1:1000 at RT) and mounted with Prolong Gold Antifade reagent with DAPI (Thermofisher, Waltham, USA). The number of CXCR3B+ immunoreactive cells (red) were counted over 10 non-overlapping fields and expressed as CXCR3B+ immuno-fluorescence per $\mu m^2$. For in situ detection of CXCR3B+ melanocytes in the human skin, OCT frozen sections were permebilised with PBS 0.3% Triton$^{TM}$ X-100 (Abcam, France) for 10 min prior to saturation with PBS 5% BSA 10% normal goat serum and 0.05% Triton$^{TM}$ X-100 for 2 h at RT. Slides were then incubated overnight at 4 °C with primary antibodies (polylonal anti-MITF, 1:50, Sigma Aldrich and monoclonal anti-CXCR3B, 1:50, Proteintech) and the next day 1 h with secondary antibodies (anti-mouse AF594 and anti-rabbit AF488, both 1:1000) prior to mounting the slides and visualizing under confocal microscope (Nikon A1R using a ×60 objective. For each slide (5 subjects per group), the number of MITF+CXCR3B+ cells (yellow immunostaining) were counted and averaged over six non-overlapping fields. CXCR3B staining in melanocytes was confirmed using another melanocyte marker gp100 (PMEL, premelanosome protein, 1:200, Abcam).

**Live imaging of melanocyte viability and proliferation.** Melanocytes were plated in 96-well plates (10,000/well) and stimulated with 50 ng/ml IFNγ for 48 h prior to transfection with 50 or 80 nM siCXCR3 or siC (SmartPool scramble sequence siRNA, Dharmacon, France) or with 80 nM custom-designed siCXCR3B or siC (ThermoFisher, France) using Lipofectamine® RNAiMAX Reagent (Invitrogen, France) in optiMEM medium (Invitrogen). Next day, media was replaced and melanocytes stained with CellTracker® Red CMPTX dye for 20 min at 37 °C (1 μM, Molecular Probes, USA) before addition of 100 pg/ml recombinant human CXCL9, CXCL10 or CXCL11 (PeproTech) to melanocytes. Finally, Incucyte® green Cytotox Reagent (100 nM, Essen Bioscience, Michigan, USA) was added to all wells and melanocyte death monitored in real-time using IncuCyte® Zoom live-cell imaging system (Essen Biosciences) which was inside a 37 °C humidified $CO_2$ incubator scanning the plate every 2 h. Multiple images were collected per well and quanti-fication of dead melanocytes (yellow co-localised cells) was analysed using the integrated Zoom® software. In separate experiments, non-transfected cells were pre-incubated with IFNγ and 48 h later treated with CXCR3 antagonist AS612568 (0.02, 0.2, 2 μM, Calbiochem, China) and Cytotox® Reagent added just before live imaging. To try and mimic an in vivo situation, vitiligo melanocytes were stimu-lated with IFNγ (to upregulate their CD40 and CXCR3B expression) and 48 h later stimulated with CXCL10 (100 pg/ml). The next day, supernatant was removed and replaced with fresh media. In separate experiments, 24 h post CXCL10 stimulation, all media was replaced and melanocytes left in culture for another 72 h before adding either allogeneic PBMC (positive control experiments) or autologous non-stimulated and sorted CD3+ T cells (1 × 10⁶/ml) prior to IncuCyte live imaging. At the end of the experiment (~40 h later), remaining melanocytes were trypsinized and co-stimulatory (CD40, HLA-DR, CD80) and adhesion cell markers (ICAM-1) on melanocytes, as well as proliferating T cells (CD3+ and Ki67+) were examined by immunofluorescence staining of cytospin sections. Monoclonal antibodies directed against human CD40 (G28.5, 1:100), HLA-DR (TU36, 1:200), CD80 (L307.4, 1:50) and ICAM-1 (HA58, 1:400) were purchased from BD Biosciences (San Diego, CA, USA). Ki67 rabbit monoclonal antibody (SP6, 1:200) was pur-chased from Abcam (Cambridge, UK). The number of CD3+ T cells in IncuCyte was counted from time-lapse images with Fiji software using a macro whereby a median filter was initially applied prior to application of 'Find Maxima' function to identify and count all non-stained dark spots in the images. Quantification of proliferation was performed using flow cytometry and CellTrace CFSE cell pro-liferation kit (Thermo Fisher Scientific, Illkirch, France). Briefly, 1 × 10⁶ cells were labelled with 5 μM of CFSE in 96-well plates and 72 h later, cells collected, stained with anti-CD3 conjugated to APC (BD Biosciences, 1:100) and fluorescence measured with MACSQuant Analyzer (Miltenyi, Paris, France). Labelled cells at time zero was used as a negative reference, unstimulated cells left in culture for 72 h before labelling as a control and cells stimulated with PHA for 72 h (Phytohe-magglutinin, 5 μg/ml) as a positive control.

**CXCL10-induced signalling in healthy and vitiligo melanocytes.** Protein sam-ples from primary melanocytes stimulated with CXCL9, CXCL10, CXCL11 (100 pg/ml) or IFNγ (50 ng/ml) for 24 h were extracted in buffer containing 50 mmlo/L Tris-HCl (pH 7.5), 15 mmol/L NaCl, 1% Triton X-100 and 1× protease and phosphatase inhibitors. Cell lysates (30 mg) were separated on SDS-polyacrylamide gel (SDS-PAGE) and transferred onto a polyvinylidene difluoride membrane (Millipore Corp). The membranes were incubated with antibodies directed against total and active p38 and poly ADP-ribose polymersase (PARP) (Cell Signalling Technology, 1:1000) followed by peroxidase-linked secondary antibodies. HSP90 was used as house-keeping genes. Reactive protein bands were detected using chemiluminescent substrate (ECL System, Amersham) and Fujifilm LAS-4000 system. Staurosporine (1 μg/ml) was used as a positive control. In parallel experiments, to confirm the role of apoptosis in CXCL10-induced melanocyte death, cells were pretreated with IFNγ and stimulated with CXCL10 (100 pg/ml) in presence of absence of QVd OPh (10 μM, caspase inhibitor, Biosciences) for 24 h. Cell lysates were prepared for assessment of total and cleaved Caspase-3 activity (Cell Signalling Technology, 1:1000) by WB and remaining cells stained with Annexin V conjugate (Miltenyi, France) and DAPI for FACS analysis.

**Statistical analyses.** Statistical analyses were performed with Graphpad Prism® 6 software. Mann-Whitney non-parametric analysis was used to test unpaired differences between groups and Wilcoxon signed rank test for paired differences. Differences were considered significant at $P < 0.05$.

## Data availability
The data that support the findings of this study are available from the corresponding author upon reasonable request.

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

## Acknowledgements

We gratefully acknowledge the Centre Méditerranéen de Médecine Moléculaire (C3M INSERM U1065) microscopy and imaging facility, particularly the expertise of platform manager Maéva Gesson. IncuCyte was available with financial support from ITMO Cancer Aviesan (Alliance National pour les Sciences de la Vie et de la Sante, National Alliance for Life Science and Health) within the framework of the Cancer Plan. Special acknowledgements to Sonia Amroune and Raja Bahroumi for their assistance in the clinic with patient recruitment. This work was supported by the Institut National de la Santé et de la Recherche Médicale (INSERM).

## Author Contributions

M.K.T. conceived and performed the experiments and wrote the manuscript. E.C. was a Master 2 student on this project. Y.C., A.J. and C.L. assisted with cell sorting and FACS analyses and provided intellectual input. N.C.L. performed and interpreted IHC analysis. P.A., H.H.B., L.S. and C.L. performed the Western blot analyses. C.R., L.S., M.G., G.E.B. and C.P. provided technical assistance and support. A.K. helped collecting patient and control samples. Y.C., A.J., C.L., R.B., C.B. and S.R. provided expertise and feedback on manuscript draft. T.P. conceived experiments, supervised the study and wrote the manuscript. All authors have read and approved the final version of the manuscript.
