## [Peer Review File · Nature Communications]

Reviewers' comments:

Reviewer #1 (Remarks to the Author):

Overall, this manuscript contains intriguing and novel data that offer a potential mechanisms by which apoptosis is induced in melanocytic cells in vitiligo. However, the obtained data, although providing multiple clues, raised several concerns regarding interpretation of the data.

Major point:

1. The title of the manuscript is somewhat misleading. It reflects the most intriguing data of the study, but does not reflect all studies regarding NK cells.

2. On the contrary to the title, the introduction of the manuscript is mostly focused on NK cells with very little information describing melanocytes.

3. Most of authors conclusions are based on the statement that observed abnormalities occur in non-lesional skin of vitiligo patients. However, it is not stated at what sites of the body these non-lesional skin biopsies were collected. Was these samples collected at perilesional skin? Was it sun-exposed or sun-protected skin? Considering that patients with stable disease were recruited to the study (no progression of existing lesions and no new lesions within 1 year) it is somewhat unexpected to detect described "abnormalities" at random skin sites.

4. Fig. 1 illustrates the presence of IFN gamma-producing NK cells in non-lesional skin.

a) Although authors showed a single cell image on micrographs of immunofluorescence, a broader field is necessary.

b) Individual NK cells detected by immuno-histochemical staining cannot be describe as elevated infiltration of the skin with CD56+ NK or ILC1 cells.

c) all presented stainings should include control healthy skin samples

5. Fig. 2 illustrates response of the NK cells and melanocytes to oxidative stress.

a) Fig 2d showed that control and vitiligo melanocytes express somewhat similar amount of the chemokines as assessed on mRNA level. At the same time, secretion of the two of them from vitiligo cells is several times higher than from control melanocytes. How this phenomenon could be explained?

b) Fig. 2f shows that melanocytes secrete CXCL9, CXCL10 and IFN gamma when co-cultured with hydrogen peroxide-treated NK and/or ILC cells. It is not clear, what is the contribution of the NK/ILC cells to the reported levels? Do melanocytes secrete IFN gamma. If these are a co-culture experiments, an additional control is required (stimulated NK/ILC alone).

6. Data showing expression of CXCR3B isoform in melanocytes and its induction by IFN gamma is very interesting (Fig. 3). Induction of apoptosis in vitiligo melanocytes after exposure to IFN gamma and CXCL10 is also exciting.

a) However, considering current title the manuscript, there is very little discussion of different functional activities of the two CXCR3 isoforms. For example, it was shown that CXCR3 agonists bound CXCR3-A with higher affinity than CXCR3-B and that lower concentrations (≤ 1 nM) of CXCR3 inhibitors are more efficiently inhibited CXCR3-A than CXCR3-B. Considering different level of expression of CXCR3B and different response of control and vitiligo melanocytes to CXCL10, it is important to show, how normal melanocytes respond to CXCL10 in the presence of CXCR3 antagonists.

b) Minor - Fig. 3a – change from baseline – is it fold difference?

General comments:

7. Prior studies demonstrated that IFN-gamma production by NK cells required the presence of CD4+ T cells or IL-2 at peripheral sites. In fact, even images provided by the manuscript somewhat support this notion: in Fig. 1 there are several IFN-gamma+ non-NK cells per single IFN-gamma+ NK cell. In addition, infiltration of tissues with NK cells (3-5 cells per section) is substantially lower than it was previously reported for T cells in non-lesional skin of vitiligo patients and healthy individuals. Out of all innate immune cells, eosinophils are most frequently observed in vitiligo lesional and perilesional skin. Therefore, it is likely, the statement that NK cells are the first to trigger IFN-gamma production, secretion of CXCL10 and apoptosis in melanocytes could be misleading.

8. Overall, it appears that phenotypically normal vitiligo melanocytes isolated from the unaffected non-lesional skin respond to increased levels of IFN gamma by apoptosis. This means that any inflammatory skin reaction could trigger apoptosis at any random site of the body and progression of vitiligo. It might be true for universal vitiligo. Can the authors explain why patients with stable vitiligo do not develop new lesions for a relatively long period?

9. Based on presented data it appear that both phenotypically normal melanocytes as well as NK and ICL cells from vitiligo patients differently respond to various stimuli when compared to cells from healthy donors. That means that all these cells harbor genetic abnormalities. Indeed, vitiligo genetic susceptibility markers were extensively studies and characterized. However, there is no discussion of this phenomenon.

Minor points:

1. First sentence in "NK and ILC1 populations are increased in vitiligo skin and blood" the authors stated that they: "...used specific antibodies directed against NK and ILC1 subpopulation... and examined expression in vitiligo patients..." or in the second paragraph "...expression of innate immune cells..." and in many other places throughout the manuscript, the authors inappropriately use word "expression" when describing presence of cells.
2. Fig 2a – please, mark x-axis.

Reviewer #2 (Remarks to the Author):

This manuscript investigates a potential role for CXCR3B on melanocytes in the development of vitiligo. The paper is well written with some novel insights into melanocyte biology but some conclusions would be better supported by the inclusion of further data particularly confirming the role of the CXCR3B isoform and T cells in the model (see comments below). The following specific points are made to help improve the manuscript:

- (1) Figure 1 is poorly presented. Fig. 1a - Is "VIIS- " three different patients? Fig. 1b – is this stained with both CD3 and CD56? More detail should be provided in the methods and figure legend about the antibodies used and how the staining was performed for Fig. 1a and b. Fig. 1e (and in other figures) – There are very few cells here being compared (average 2 cells vs 0.5 cell) and so it is a little surprising that the error bars are so tight. To increase the sensitivity of detection, a flow cytometry experiment on fresh, digested tissue with quantitation would complement this microscopic data and improve the paper.
- (2) Fig. 2a is missing X-axis labels.
- (3) Fig. 4b and Supp Fig. 2 - Is the siCXCR3 knockdown specific for CXCR3B? This would seem critical to the title of the paper. What about the effect on the other CXCR3 isoforms?
- (4) Fig. 4d shows cleaved PARP and p38 for CXCL10 treatment but is this a concentration dependent effect and CXCL10 is just more efficient at this dose than CXCL9 and CXCL11? Have CXCL9 and 11 been used in the melanocyte cell death assays?
- (5) Fig. 5 – The authors need to rule out a direct effect of CXCL10 on the T cells (perhaps through residual CXCL10 in the co-culture). What would happen if the T cells were pre-incubated with CXCL10 and then placed in co-culture with IFN-g treated melanocytes? In addition, the microscopic quantitation should be supported by flow cytometric quantitation of proliferation and T cells for the whole culture in Fig. 5c and d.
- (6) Fig. 6 – There is no evidence presented in the manuscript to suggest that naive, CD4 T cells are responding to melanocytes. This would require a full characterisation of the responding T cells in Fig. 5. It would also need to be demonstrated that naive, CD4 T cells are capable of migrating to melanocytes in the epidermis. Finally, the release of melanocytic antigens and the role this would have in stimulating T cells has not been addressed in the figures.

Reviewer #3 (Remarks to the Author):

- 1) Do NK and ILC1 cells produce CXCL8, CXCL10 or CXCL11? This is important because your model predicts that CXCL10 may be responsible for the death of melanocytes during vitiligo. However, if the NK and ILC1 cells that you propose as the initial IFN γ -producing cells in asymptomatic vitiligo skin produce also CXCL9, CXCL10 (or CXCL11) then your model may need revision, because you would have CXCL9, CXCL10 or CXCL11 present in asymptomatic skin without melanocyte death. In fact, Immgen indicates that subpopulations of both NK and ILC1 cells can produce significant amounts of CXCL10, so this is a possibility that should be explored.
- 2) Similarly, you should perform qPCR to monitor for expression of CXCL9, CXCL10 and CXCL11 in asymptomatic (ie adjacent normal) versus symptomatic skin. If their model is correct, only the symptomatic skin should express high levels of CXCL10, specially, since this is the chemokine you show induces melanocyte death.
- 3) CXCL9, CXCL10 and CXCL11 bind CXCR3. CXCL11 is a higher affinity ligand than CXCL10 for CXCR3. What is the relationship of these ligands to CXCR3B? Do they exhibit the same affinity to CXCR3B than to CXCR3? This information must be in the literature and you should mention it. If this is the case, you may want to compare the capacity of CXCL11 to induce death of melanocytes to CXCL10. You should also compare the ability of different skin cells to produce CXCL11, not only CXCL9 and CXCL10.
- 4) Why did you measure CXCL4 in Figure 2D? was it a control? What is its significance?
- 5) You show that melanocytes express CXCR3B. But in Figure 4, you use an inhibitor of CXCR3 to show that CXCL10 induces death of melanocytes. Has the ability of the CXCR3 antagonist AS612568 to block CXCR3B been assessed? As presented, your data are confusing. Do you think that the effect of CXCL10 inducing death of melanocytes is mediated by CXCR3B? Or is it going through CXCR3A?
- 6) In your models, you show that melanocytes express CXCR3B but T cells express CXCR3A. But you only show data on CXCR3B on melanocytes. Did you check CXCR3B expression in T cells? It appears that you concluded that melanocytes express CXCR3B only based on the specificity of a commercial antibody. There are antibodies available that recognize human CXCR3 (presumably CXCR3A, since most CXCR3 expression in T cells should be CXCR3A). Did you confirm that melanocytes do not express CXCR3A? Or do you believe that melanocytes only express CXCR3B? Do you hypothesize that the death inducing effect of CXCL10 is only mediated through CXCR3B (and not CXCR3A)?
- 7) As listed above, some of your observations are interesting but the paper as presented needs better explanation about what you are proposing. The models help but there are significant questions. For example, if the model is correct, there should be more CXCL10 expression in lesional vs adjacent normal skin. You should perform these experiments.
- 8) Some papers in the literature have documented that CXCL9 and CXCL10 are strongly produced during some autoimmune and inflammatory diseases. This leads to accumulation of CXCR3+ T cells that show Th1 phenotypes (including IFN γ production). You should mention these papers in your discussion.

Centre Méditerranéen de Médecine Moléculaire
U1065

Reviewers' comments:

Reviewer #1 (Remarks to the Author):

Overall, this manuscript contains intriguing and novel data that offer a potential mechanisms by which apoptosis is induced in melanocytic cells in vitiligo. However, the obtained data, although providing multiple clues, raised several concerns regarding interpretation of the data.

Major point:

1. The title of the manuscript is somewhat misleading. It reflects the most intriguing data of the study, but does not reflect all studies regarding NK cells.

*_ We agree and we are proposing a new title reflecting the role of the innate immunity and the activation of CXCR3B : “**Dysregulation of innate immunity resulting in CXCR3B-mediated apoptosis of melanocytes initiates vitiligo**”*

2. On the contrary to the title, the introduction of the manuscript is mostly focused on NK cells with very little information describing melanocytes.

_ Accordingly to your recommendation, we have now added extra information regarding the potential role of melanocytes in the introduction.

3. *Most of authors conclusions are based on the statement that observed abnormalities occur in non-lesional skin of vitiligo patients. However, it is not stated at what sites of the body these non-lesional skin biopsies were collected. Was these samples collected at perilesional skin? Was it sun-exposed or sun-protected skin? Considering that patients with stable disease were recruited to the study (no progression of existing lesions and no new lesions within 1 year) it is somewhat unexpected to detect described “abnormalities” at random skin sites.*

_ The samples were collected from non-lesional skin and not from their perilesional skin. All the biopsies taken were from the sun-protected skin. We have now amended Table 1 to include sites from where the biopsies were taken.

It is true that finding an increased innate immunity in non-lesional skin might be surprising. However, we and others have already shown an increase in the innate immune cells markers, as well as CXCLs and IFN γ in non-lesional skin of vitiligo patients compared to completely depigmented skin or healthy skin (PLoS One. 2012;7(12):e51040; J Invest Dermatol. 2015 Dec;135(12):3105-3114 ; Pigment Cell Melanoma Res. 2017 Mar;30(2):259-261). In addition, we have also previously shown that CXCL10 and IFN γ are significantly higher in the non-lesional skin of active vitiligo patients compared to the non-lesional skin of stable vitiligo patients (Pigment Cell Melanoma Res. 2017 Mar;30(2):259-261). As we demonstrated using innate cells from vitiligo patients and healthy volunteers, there is a basal activation of NK and ILC1 in vitiligo skin. However, those innate cells respond much more to DAMPs leading to a strong increase in IFN γ . Thus, although abnormalities are found in non-lesional skin, a trigger is still needed to strongly increase the production of IFN γ and to initiate the auto-immune response against melanocytes at this site.

4. *Fig. 1 illustrates the presence of IFN gamma-producing NK cells in non-lesional skin. a) Although authors showed a single cell image on micrographs of immunofluorescence, a broader field is necessary.*

_ We have made a new figure 1 that include broader fields of vision.

b) *Individual NK cells detected by immuno-histochemical staining cannot be describe as elevated infiltration of the skin with CD56+ NK or ILC1 cells.*

_ In Figure 1ab we counted the number of CD56+GranzymeB+ and the number of CD56+IFN γ + cells in 10 non-overlapping regions and divided the results by number of fields counted (dependent on the length of tissue). We apologize for the error in y-axis labeling which should read CD56+GranzymeB+ cells/field or CD56+IFN γ +cells/field. This has now been rectified in Figure 1.

c) *all presented stainings should include control healthy skin samples*

_ This has now been done.

5. Fig. 2 illustrates response of the NK cells and melanocytes to oxidative stress.

a) Fig 2d showed that control and vitiligo melanocytes express somewhat similar amount of the chemokines as assessed on mRNA level. At the same time, secretion of the two of them from vitiligo cells is several times higher than from control melanocytes. How this phenomenon could be explained?

_ Measuring mRNA is a good indicator of gene regulation, but it cannot be assumed that the mRNA amount is directly correlated with protein expression since post-transcriptional processes are key to the final synthesis of the native protein. It is not unusual to see difference in mRNA and protein expression (https://kendricklabs.com/wp-content/uploads/2016/08/WP1_mRNAsvsProtein_KendrickLabs.pdf). For chemokines that are secreted, the difference between mRNA expression and protein detected in the supernatant can be explained not only by post-transcriptional processes but also by the secretion of proteins by the cells. It is for this reason we have chosen to measure chemokines by ELISA in the supernatant.

b) Fig. 2f shows that melanocytes secrete CXCL9, CXCL10 and IFN gamma when co-cultured with hydrogen peroxide-treated NK and/or ILC cells. It is not clear, what is the contribution of the NK/ILC cells to the reported levels? Do melanocytes secrete IFN gamma. If these are a co-culture experiments, an additional control is required (stimulated NK/ILC alone).

_ We apologize if this was not clearly explained in the text. In Figure 2f, yes we have examined separately the contribution of NKs or ILCs or NKs+ILCs on melanocyte production of CXCL9, CXCL10 or IFN γ (we have now also added CXCL11 following other reviewers comments). To answer the question about melanocytes production of IFN γ , we have measured constitutive levels in non-stimulated vitiligo melanocytes (48 ± 15 pg/ml) and have shown these levels to be similar to those from melanocytes from healthy controls (27 ± 11 pg/ml) ($P=0.34$). Results are now shown in Figure 2f graph on the bottom right (condition positive control C).

Secondly, to address the question regarding the contribution of NK/ILC to the chemokine levels we have now performed additional experiments and directly measured CXCL9, CXCL10 and CXCL11 from the original supernatants in Exp 2a where we looked at time course of IFN γ production in NK and ILC following H₂O₂ stimulation (Figure 2). Further experiments were also performed looking at stimulation with other innate stimuli such as HMGB1 and HSP70 (Supplementary Figure 2). These new data have shown that NK/ILC production of CXCL9, CXCL10 and CXCL11 following innate stress is negligible compared to the NK/ILC production of IFN γ following the same stress stimuli and that this chemokine production is also negligible compared to the production by melanocytes (Figure 2f). These additional data suggest that the major producers of chemokines following stress in our model are the melanocytes and not the NK or ILCs. These new data have now been added to the Results (Supplementary Figure 2).

6. Data showing expression of CXCR3B isoform in melanocytes and its induction by IFN gamma is very interesting (Fig. 3). Induction of apoptosis in vitiligo melanocytes after exposure to IFN gamma and CXCL10 is also exciting.

a) However, considering current title the manuscript, there is very little discussion of different functional activities of the two CXCR3 isoforms. For example, it was shown that CXCR3 agonists bound CXCR3-A with higher affinity than CXCR3-B and that lower concentrations (≤ 1 nM) of CXCR3 inhibitors are more efficiently inhibited CXCR3-A than CXCR3-B. Considering different level of expression of CXCR3B and different response of control and vitiligo melanocytes to CXCL10, it is important to show, how normal melanocytes respond to CXCL10 in the presence of CXCR3 antagonists.

_ We acknowledge the reviewer's comments and have now repeated the experiment in original Figure 4a however this time using normal healthy (and not vitiligo) melanocytes. Our results have shown a dose-dependent reduction in melanocyte death with increasing concentration of CXCR3 antagonist (amended Figure 4a). To more specifically examine the contributing role of CXCR3B, we have now repeated this experiment in both healthy and vitiligo melanocytes using custom design specific silencer RNA directed against the CXCR3B isoform and have included these new data in Figure 4d. In addition to CXCL10 (and in response to the other reviewer's comment) we have now examined responses to not only CXCL10 but to other chemokines including CXCL9 and CXCL11 (new data in Figure 4d).

b) Minor - Fig. 3a – change from baseline – is it fold difference?

_ We apologize for this oversight. Yes, it is fold difference and this has now been added to the Figure for clarification.

General comments:

7. Prior studies demonstrated that IFN-gamma production by NK cells required the presence of CD4+ T cells or IL-2 at peripheral sites. In fact, even images provided by the manuscript somewhat support this notion: in Fig. 1 there are several IFN-gamma+ non-NK cells per single IFN-gamma+ NK cell. In addition, infiltration of tissues with NK cells (3-5 cells per section) is substantially lower than it was previously reported for T cells in non-lesional skin of vitiligo patients and healthy individuals. Out of all innate immune cells, eosinophils are most frequently observed in vitiligo lesional and perilesional skin. Therefore, it is likely, the statement that NK cells are the first to trigger IFN-gamma production, secretion of CXCL10 and apoptosis in melanocytes could be misleading.

_ Virus specific T cells are required for virus induced production of IFN γ by NK (J Clin Invest. 2004 Nov; 114:1812-1819) however IFN γ can be produced by NK cells upon NKR-P1 cross-linkin, therefore no T cell involvement (J Exp Med 1996 May; 183:2391-2396). NK cells are a major innate source of IFN γ produced rapidly during many infections before the development of an adaptive response. Unlike for T cells for who cytokines alone fail to stimulate them to produce IFN γ and the interaction of the TCR with its specific Ag is a prerequisite to initiate their production of IFN γ , in contrast, NK cells can be stimulated directly with cytokines to produce high levels of IFN γ . The mechanism by which this occurs was shown by Tato and colleagues in 2004 who have revealed that resting NK cells can make IFN γ within hours of cytokine stimulation, but these events are independent of proliferation (J Immunol. 2004; 173:1514-1517) . Moreover, they have shown that IFN γ locus in resting NK cells, in contrast to naive T cells, appears to exist in a more open configuration that correlates with the ability of

these cells to make IFN γ rapidly. Their findings reveal that differences in the structure of the IFN γ gene provide a molecular basis for the different kinetics of innate and adaptive IFN γ responses (J Immunol. 2004; 173:1514-1517).

The reviewer mentioned that “eosinophils are most frequently observed in vitiligo lesional and perilesional skin”. However, our study focused on the non-lesional vitiligo skin to decipher the initial mechanisms involved. Moreover, to the best of our knowledge there is very little data in the literature regarding the role of eosinophils in vitiligo. Some reports suggest that there is no eosinophils in vitiligo (Eur J Haematol. 1988 Apr;40(4):368-70). We found only one retrospective study showing the presence of eosinophils in only 10% of the studied patients and only in active cases (Clin Exp Dermatol. 2009 Oct;34(7):e496-7).

8. Overall, it appears that phenotypically normal vitiligo melanocytes isolated from the unaffected non-lesional skin respond to increased levels of IFN gamma by apoptosis. This means that any inflammatory skin reaction could trigger apoptosis at any random site of the body and progression of vitiligo. It might be true for universal vitiligo. Can the authors explain why patients with stable vitiligo do not develop new lesions for a relatively long period?

_ This is a very interesting question. So far, despite a better understanding of the adaptive immune processes involved in vitiligo depigmentation, there is almost no data to explain the development of new lesions and why, when the depigmentation process is engaged, it remains localized to some area of the body (and then sometimes remains stable for years). We believe that our results provide the first answers to these phenomena. We demonstrated that there is a basal activation of NK and ILC1 in the skin and the blood of vitiligo patients. We also showed that those innate cells respond more to DAMPs stimulation, inducing a strong production of IFN γ . This local secretion of IFN γ induces the production of CXCL10 by keratinocytes and by melanocytes. Interestingly, we showed that there is a dose-response production of IFN γ by the innate cells after stimulation by DAMPs (Figures 2a, b and c) and in response, an increased production of chemokines by the melanocytes (Figure 2f). Moreover, we also showed a dose response apoptosis of melanocytes when treated with increasing amounts of CXCL10 (Figure 4a). Thus, we can hypothesize that in the basal state in non-lesional vitiligo skin, the increased production of CXCL10 and IFN γ is not sufficient to induce melanocyte apoptosis and the attraction of lymphocytes. Upon local stress (physical or chemical injuries, bacteria, virus...), the innate cells react and produce higher amounts of IFN γ . We can hypothesize that above a certain threshold, the local production of IFN γ triggers the downstream processes that we described. We have now made a new scheme trying to better summarize these findings (Figure 6).

9. Based on presented data it appear that both phenotypically normal melanocytes as well as NK and ICL cells from vitiligo patients differently respond to various stimuli when compared to cells from healthy donors. That means that all these cells harbor genetic abnormalities. Indeed, vitiligo genetic susceptibility markers were extensively studied and characterized. However, there is no discussion of this phenomenon.

_ Thank you. This is a good point. We have further discussed this in the revised version of the manuscript.

Minor points:

1. First sentence in “NK and ILC1 populations are increased in vitiligo skin and blood” the authors stated that they: “...used specific antibodies directed against NK and ILC1 subpopulation... and examined expression in vitiligo patients... “or in the second paragraph “...expression of innate immune cells...” and in many other places throughout the manuscript, the authors inappropriately use word “expression” when describing presence of cells.

_ We changed this throughout the manuscript.

2. Fig 2a – please, mark x-axis.

Thank you for pointing this out. It has now been rectified.

Reviewer #2 (Remarks to the Author):

This manuscript investigates a potential role for CXCR3B on melanocytes in the development of vitiligo. The paper is well written with some novel insights into melanocyte biology but some conclusions would be better supported by the inclusion of further data particularly confirming the role of the CXCR3B isoform and T cells in the model (see comments below). The following specific points are made to help improve the manuscript:

_ We thank the reviewer for the positive feedback regarding our study. We have now performed a number of additional experiments which we hope to have answered all your concerns below.

(1) Figure 1 is poorly presented. Fig. 1a - Is “VIIS- “ three different patients? Fig. 1b – is this stained with both CD3 and CD56? More detail should be provided in the methods and figure legend about the antibodies used and how the staining was performed for Fig. 1a and b. Fig. 1e (and in other figures) – There are very few cells here being compared (average 2 cells vs 0.5 cell) and so it is a little surprising that the error bars are so tight. To increase the sensitivity of detection, a flow cytometry experiment on fresh, digested tissue with quantitation would complement this microscopic data and improve the paper.

_ We put the same patient pictures in the figure to better show the difference between the staining in lesional and non-lesional skin. Of course, the healthy skin control comes from a distinct control patient. We have now included broader field images and explained in more detail the staining protocol used. Unfortunately we couldn't perform new flow cytometry experiments in fresh skin as it would require to submit a new authorization to include additional patients in order to get new fresh skin samples. However, flow cytometry was already performed in the blood of the patients. Finally, although the number of positive cells are quite low (which is expected for NKs and ILCs) the difference between non-lesional, lesional and control skin samples was very consistent explaining the tight error bars and the strong statistical significances detected.

(2) Fig. 2a is missing X-axis labels.

_ Thank you for pointing this out. It has now been rectified.

(3) Fig. 4b and Supp Fig. 2 - Is the siCXCR3 knockdown specific for CXCR3B? This would seem critical to the title of the paper. What about the effect on the other CXCR3 isoforms?

_ The original siCXCR3 was not specific for CXCR3B isoform however although we have knocked down total protein we have shown that when we used CXCR3B antibody we inhibited ~ 80% of the CXCR3B mRNA and protein in normal human melanocytes (now Supplementary Figure 3ab). We have however taken the reviewers comments on board and have custom designed siCXCR3 knockdown specific for the CXCR3B isoform (confirmation of its specificity shown in supplementary Figure 3c). The key experiments have been redone with selective siCXCR3B using not only CXCL10 chemokine but also other CXCR3B agonists including CXCL9 and CXCL11 in response to further questions below (Figure 4ad).

(4) Fig. 4d shows cleaved PARP and p38 for CXCL10 treatment but is this a concentration dependent effect and CXCL10 is just more efficient at this dose than CXCL9 and CXCL11? Have CXCL9 and 11 been used in the melanocyte cell death assays?

_ This is a valid comment by the reviewer and we have now performed additional melanocyte cell death assays to answer this question. Importantly, these experiments have now been performed using siRNA specific for the CXCR3B isoform (new data added in Figure 4d). These results clearly show that indeed, CXCL9 and CXCL11 do induce melanocyte death and this is driven by CXCR3B. However, the comparison to CXCL10-induced death shows 2-3 and 7 fold difference for CXCL9 and CXCL11, respectively. These results are intuitive as we can imagine that the selectivity for CXCL10 is not 100% and that CXCL9 and 11 could partially and less efficiently activate the CXCR3B receptor. It has previously been shown that CXCR3 splice variants activate different signaling pathways, suggesting a mechanism for tissue specific biased agonism such as that reported for CXCR3A in prostate cancer (Mol Cancer. 2012 Jan; 11:3 doi: 10.1186/1476-4598-11-3), CXCR3B in breast cancer (J Biol Chem. 2014 Feb; 289(6):3126-3137) or a role for CXCR3Alt in Crohn's disease (J Gastroenterol Hepatol 2008 Dec; 23(12):1823-1833). These important new results add supporting data to our paper are now discussed in the manuscript.

(5) Fig. 5 – The authors need to rule out a direct effect of CXCL10 on the T cells (perhaps through residual CXCL10 in the co-culture). What would happen if the T cells were pre-incubated with CXCL10 and then placed in co-culture with IFN-g treated melanocytes? In addition, the microscopic quantitation should be supported by flow cytometric quantitation of proliferation and T cells for the whole culture in Fig. 5c and d.

_ To answer this question, we have recalled existing vitiligo patients from our database whose melanocytes we had previously extracted and have taken fresh blood samples from them to perform co-culture experiments. As reviewer suggested, we have sorted CD3+ T cells from PBMC of vitiligo patients and pre-incubated them with CXCL10 for 24hrs, prior to their addition to IFN γ pretreated melanocytes. Our results have shown that prior incubation of T cells with CXCL10 prior to co-culture with IFN γ -treated melanocytes has no significant effect on melanocyte death (new data added as supplementary Figure 6). In addition, we have performed flow cytometry analysis to examine and quantitate the proliferation of T cells when they were pre-incubated with CXCL10 or cultured with melanocytes (in conditions with or without IFN γ pre-stimulation). The number of cells divisions between treatment groups was assessed by CFSE staining. Results have shown T cells co-cultured with melanocytes pre-treated with IFN γ and CXCL10 proliferated to the greater extent than those T cells which were pretreated with CXCL10 and then incubated with IFN γ -exposed melanocytes. This proliferation was blocked by addition of CXCR3 antagonist (new data added to Figure 5e).

(6) Fig. 6 – There is no evidence presented in the manuscript to suggest that naïve, CD4 T cells are responding to melanocytes. This would require a full characterization of the responding T cells in Fig. 5. It would also need to be demonstrated that naïve, CD4 T cells are capable of migrating to melanocytes in the epidermis. Finally, the release of melanocytic

antigens and the role this would have in stimulating T cells has not been addressed in the figures.

This was not the main focus of the paper and we have therefore now reduced the emphasis on these results for the risk of over-interpretation of the role of naïve T cells response to melanocytes. Taking into consideration the comments of all the reviewers, we are proposing a new Figure 6 focusing only on the results obtained in the present study.

Reviewer #3 (Remarks to the Author):

1) Do NK and ILC1 cells produce CXCL9, CXCL10 or CXCL11? This is important because your model predicts that CXCL10 may be responsible for the death of melanocytes during vitiligo. However, if the NK and ILC1 cells that you propose as the initial IFN γ -producing cells in asymptomatic vitiligo skin produce also CXCL9, CXCL10 (or CXCL11) then your model may need revision, because you would have CXCL9, CXCL10 or CXCL11 present in asymptomatic skin without melanocyte death. In fact, Immgen indicates that subpopulations of both NK and ILC1 cells can produce significant amounts of CXCL10, so this is a possibility that should be explored.

_ This is a really interesting remark and we have now performed additional experiments to address these comments. We have now measured CXCL9, CXCL10 and CXCL11 production from culture supernatants with ELISA using sorted NK and ILCs from healthy volunteers and vitiligo patients described in Figure 2a-c. Our results have clearly shown that in our system, NK and ILCs produce very little, if any chemokines themselves following DAMP stimulation (HMGB1 or HSP70) compared to melanocytes (Supplementary Figure 2). In contrast, NK and ILCs produce significant amount of IFN γ following stress (Supplementary Figure 2). These new data have now been added to the results and their implications acknowledged in the Discussion.

2) Similarly, you should perform qPCR to monitor for expression of CXCL9, CXCL10 and CXCL11 in asymptomatic (ie adjacent normal) versus symptomatic skin. If their model is correct, only the symptomatic skin should express high levels of CXCL10, specially, since this is the chemokine you show induces melanocyte death.

_ We and other have already shown an increase in innate cells markers, but also CXCLs and IFN γ in non-lesional skin of vitiligo patients as compared to completely depigmented skin or the skin of healthy individuals (PLoS One. 2012;7(12):e51040; J Invest Dermatol. 2015 Dec;135(12):3105-3114 ; Pigment Cell Melanoma Res. 2017 Mar;30(2):259-261). However, we have also previously shown that CXCL10 and IFN γ are significantly higher in the non-lesional skin of active vitiligo patients compared to the non-lesional skin of stable vitiligo patients (Pigment Cell Melanoma Res. 2017 Mar;30(2):259-261). As we demonstrated using innate cells from vitiligo patients and healthy volunteers, there is a basal activation of NK and ILC1 in vitiligo skin. However, those innate cells also respond much more to DAMPs leading to a strong increase of IFN γ . Thus, although abnormalities are found in non-lesional skin, a trigger is still needed to strongly increase the production of IFN γ and to initiate the auto-immune response against melanocytes. Thus, we showed that there is a dose response production of IFN γ by the innate cells after DAMPs stimulation (Figures 2a, b and c) and in response, an increased production of chemokines by the melanocytes (Figure 2f). Moreover, we also showed a dose response apoptosis of melanocytes when treated with increasing amounts of CXCL10 (Figure 4a). Thus, we can hypothesize that in the basal state in non-lesional vitiligo skin, the increased production of CXCL10 and IFN γ is not sufficient to induce melanocyte apoptosis and the subsequent attraction of lymphocytes. Upon local stress (physical or chemical injuries, bacteria, virus), the innate cells react and produce higher amounts of IFN γ . We can hypothesize that above a threshold, the local production of IFN γ triggers the downstream processes described in this paper. We made a new scheme trying to better summarize these findings.

3) *CXCL9, CXCL10 and CXCL11 bind CXCR3. CXCL11 is a higher affinity ligand than CXCL10 for CXCR3. What is the relationship of these ligands to CXCR3B? Do they exhibit the same affinity to CXCR3B than to CXCR3? This information must be in the literature and you should mention it. If this is the case, you may want to compare the capacity of CXCL11 to induce death of melanocytes to CXCL10. You should also compare the ability of different skin cells to produce CXCL11, not only CXCL9 and CXCL10.*

_ In the literature there does exist difference in affinity of different chemokines for the CXCR3 receptor with CXCL11 showing higher affinity than CXCL10 or CXCL9 however little information is available on the binding affinity for CXCR3 variants (reviewed in Front Med 2018 Sept; 5:271). Therefore, in response to reviewers comments, we have gone on to design and order a specific knockdown for the B isoform of the CXCR3 (siCXCR3B). In these new experiments we have looked at the capacity of not only CXCL9 and CXCL10 to induce cell death but as suggested, CXCL11. Our results have shown that although some responses are seen with CXCL9 and CXCL11, the strongest chemokine-induced death is seen in melanocytes exposed to CXCL10. In all cases, siCXCR3B was effective in reversing the chemokine-induced death of melanocytes. These 3 new experiments have now been inserted in Figure 4d showing higher sensitivity of melanocytes to CXCL10-induced death compared to CXCL9 or CXCL11 (responses to CXCL10 are 2-3 and 7-fold higher compared to CXCL9 and CXCL11, respectively).

4) *Why did you measure CXCL4 in Figure 2D? was it a control? What is its significance?*

_ We used CXCL4 as it has been previously reported to be the most important CXCR3B ligand (Immunology 2011 Apr; 132(4):503-515.). However, we confirmed that the production in the skin is negligible compared to CXCL9, 10 and 11.

5) *You show that melanocytes express CXCR3B. But in Figure 4, you use an inhibitor of CXCR3 to show that CXCL10 induces death of melanocytes. Has the ability of the CXCR3 antagonist AS612568 to block CXCR3B been assessed? As presented, your data are confusing. Do you think that the effect of CXCL10 inducing death of melanocytes is mediated by CXCR3B? Or is it going through CXCR3A?*

_ We apologize for not being clear in Figure 4. We used the CXCR3 antagonist (not specific for CXCR3B) to show that CXCL10-induced melanocyte death is driven through CXCR3. This has now been modified in the results. To answer the second question regarding the specificity of this death response to CXCR3B variant we have now designed and tested the specific siCXCR3B. Our new data (Figure 4d) show that indeed, the CXCL10 induces melanocyte death (as well as other chemokines CXCL9 and CXCL11 but to a much lesser extent) through the CXCR3B isoform which we have shown to be present in human healthy melanocytes but elevated in vitiligo melanocytes and whose expression is inducible by IFN γ (Figure 3).

6) *In your models, you show that melanocytes express CXCR3B but T cells express CXCR3A. But you only show data on CXCR3B on melanocytes. Did you check CXCR3B expression in T*

cells? It appears that you concluded that melanocytes express CXCR3B only based on the specificity of a commercial antibody. There are antibodies available that recognize human CXCR3 (presumably CXCR3A, since most CXCR3 expression in T cells should be CXCR3A). Did you confirm that melanocytes do not express CXCR3A? Or do you believe that melanocytes only express CXCR3B? Do you hypothesize that the death inducing effect of CXCL10 is only mediated through CXCR3B (and not CXCR3A)?

_ To date, tissue-specific expression of CXCR3 alternative variants has been mainly quantified using PCR as variant-specific antibodies for CXCR3A remain unavailable. It sounds logical as CXCR3B only differs from CXCR3A by an additional extracellular part. Thus, it is possible to design primers or antibodies recognizing specifically the peptides of this additional extracellular part but any primer or antibody directed against the rest of the protein will recognize both CXCR3A and CXCR3B. Using antibodies directed against the total CXCR3 and the specific CXCR3B isoform, we have shown using flow cytometry analysis that CXCR3B+CD4+ or CXCR3B+CD8+ peripheral T cells make up only a small percentage of all CXCR3total T cells (Supplementary Figure 4a). Importantly, following CXCL10 stimulation, these T cells do not undergo cell death but they proliferate (Supplementary Figure 4b) suggesting they are likely to express the CXCR3A variant which is known in the literature to play a critical role in cell chemotaxis and proliferation (Front Med. 2018 Sep; 5:271). The second comment regarding the specific expression of CXCR3B on melanocytes, ideally we would use anti-CXCR3A to show its absence however the suggested use of anti-CXCR3 is known to pick up all variants and hence it is not a reliable tool. Finally, yes we do believe that CXCL10-mediated melanocyte death is mediated through CXCR3B which was based on a number of parallel observations but in response to reviewers comments we have now repeated the key experiments with custom designed siCXCR3B with which we have now shown specific and potent inhibition of death and not only with one chemokine but also CXCL9 and CXCL11 (although with a lower sensitivity). In light of the significant amount of new data we have redrawn our model to include the combined results of our study in the visual form (Figure 6).

7) As listed above, some of your observations are interesting but the paper as presented needs better explanation about what you are proposing. The models help but there are significant questions. For example, if the model is correct, there should be more CXCL10 expression in lesional vs adjacent normal skin. You should perform these experiments.

_ In fact, we have already performed a study comparing the expression of CXCL10 in lesional, peri-lesional and non lesional vitiligo skin and we published the answer to this question (J Invest Dermatol. 2015 Dec;135(12):3105-3114; Pigment Cell Melanoma Res. 2017 Mar;30(2):259-261). Interestingly, there is a significant increase in CXCL10 in peri-lesional and non lesional vitiligo skin compared to lesional skin. However, there is no difference between peri-lesional and non lesional skin. Interestingly, we also showed that IFN γ and CXCL10 production was higher in non-lesional and peri-lesional skin of active vitiligo patients compared to patients with stable disease (Pigment Cell Melanoma Res. 2017 Mar;30(2):259-261). In lesional skin where the depigmentation is complete and all the melanocytes are lost, it is well demonstrated that there is no sign of any immune infiltrate (innate or adaptative). As we have shown in the current manuscript, the innate immune reaction occurs at the early stages and triggers the secondary adaptative immunity. As discussed above, despite a basal increase in innate immunity in non-lesional skin of vitiligo patients, this basal increase is not sufficient to induce melanocyte apoptosis and to stimulate the adaptative immune response. However, we also showed that NK

and ILC1 cells of these patients respond to a greater degree to DAMPs leading to a dose response production of IFN γ by those cells followed by a dose reponse secretion of chemokines by melanocytes. As discussed in the response to the question by Reviewer 1, these mechanisms are likely to explain why vitiligo lesions often remain restricted to certain locations in the body and are not widespread throughout the body when the auto-immune process against melanocytes is initiated. Our results help us to understand how, in perhaps genetically predisposed patients with dysregulated innate immunity and melanocyte function, local activation by DAMPs may trigger the initial immune reaction and probably explains why lesions remain localized. We have now made a new Figure 6 taking into account all the comments and the news experiments we have performed that summarize our results and the new pathophysiological insights they provide.

8) Some papers in the literature have documented that CXCL9 and CXCL10 are strongly produced during some autoimmune and inflammatory diseases. This leads to accumulation of CXCR3+ T cells that show Th1 phenotypes (including IFN γ production). You should mention these papers in your discussion.

Following your recommendations we have now added a recent review on this topic (Front Med (Lausanne). 2018 Sep 25;5:271) and further discussed this point in the manuscript.

Sincerely,

Pr. Thierry Passeron, MD, PhD

Department of Dermatology

Hôpital Archet 2

150 Route de Saint-Antoine de Ginestière

06200 Nice, France

Phone: +33 4 92 03 64 88 / Fax: +33 4 92 03 65 60

Email: passeron@unice.fr

REVIEWERS' COMMENTS:

Reviewer #1 (Remarks to the Author):

The authors critically reviewed the manuscript to address issues and concerns raised by the reviewers. New data was added to the revised manuscript. It provided additional proof for the proposed mechanism of interaction between innate immunity and melanocytes in vitiligo. Although the data demonstrating elevated infiltration of the unaffected skin of vitiligo patients with NK cells is not convincing (only few cells per field are detected), it is somewhat consistent with prior studies (Sinha, 1997). Overall, the presented data would be of interest to researchers and clinicians in cutaneous biology and pigment cell research fields.

Reviewer #2 (Remarks to the Author):

The authors have now addressed my concerns in a revised version of their manuscript.

Reviewer #3 (Remarks to the Author):

Your manuscript is much improved. However, an important conclusion is that CXCL10 exhibits special characteristics in melanoma (vs CXCL9 or CXCL11, other CXCR3 ligands). This can either be as you suggest because of expression of CXCR3b, or, alternatively, because of special properties of CXCL10. In regards to the latter possibility, you may want to discuss whether CXCL10 exhibits structural features that are unique to it and not found in CXCL9 or CXCL11.

Centre Méditerranéen de Médecine Moléculaire U1065

REVIEWERS' COMMENTS:

Reviewer #1 (Remarks to the Author):

The authors critically reviewed the manuscript to address issues and concerns raised by the reviewers. New data was added to the revised manuscript. It provided additional proof for the proposed mechanism of interaction between innate immunity and melanocytes in vitiligo. Although the data demonstrating elevated infiltration of the unaffected skin of vitiligo patients with NK cells is not convincing (only few cells per field are detected), it is somewhat consistent with prior studies (Sinha, 1997). Overall, the presented data would be of interest to researchers and clinicians in cutaneous biology and pigment cell research fields.

Response: We thank the reviewer for the positive comments regarding our revised manuscript.

Reviewer #2 (Remarks to the Author):

The authors have now addressed my concerns in a revised version of their manuscript.

Response: We thank this reviewer for the quality reviewing of our manuscript as their additional experiments requested now add further weight to our hypothesis and has improved the quality of our manuscript.

Reviewer #3 (Remarks to the Author):

Your manuscript is much improved. However, an important conclusion is that CXCL10 exhibits special characteristics in melanoma (vs CXCL9 or CXCL11, other CXCR3 ligands). This can either be as you suggest because of expression of CXCR3b, or, alternatively, because of special properties of CXCL10. In regards to the latter possibility, you may want to discuss whether CXCL10 exhibits structural features that are unique to it and not found in CXCL9 or CXCL11.

Response: In support of our results, it has previously been shown that while CXCL10 and CXCL11 have similar affinities for CXCR3A, CXCL10 has much higher binding affinity binding for CXCR3B than does CXCL9 or CXCL11 ($IC_{50} = 7$ nM versus 32 nM for CXCL11 and 133 nM for CXCL9) ¹. One potential explanation is the difference in spatial conformation between the CXCR3A and CXCR3B isoforms which plays an important role in their affinity to ligands ². We have now discussed these papers, their implications for our study and included references in support of our argument.

¹ Lasagni *et al.*, 2003 *J Exp Med* 197(11) : 1537-1549.

² Boye *et al.*, 2017 *Scientific Reports* 7 :10703/doi :10.1038/s41598-017-11151

Sincerely,

Pr. Thierry Passeron, MD, PhD
Department of Dermatology
Hôpital Archet 2
150 Route de Saint-Antoine de Ginestière
06200 Nice, France
Phone: +33 4 92 03 64 88 / Fax: +33 4 92 03 65 60
Email: passeron@unice.fr